# Active learning streamlines development of high performance catalysts for higher alcohol synthesis

Manu Suvarna [1,2], Tangsheng Zou [1,2], Sok Ho Chong[1], Yuzhen Ge[1], Antonio J. Martín [1] & Javier Pérez-Ramírez [1] ✉

Developing efficient catalysts for syngas-based higher alcohol synthesis (HAS) remains a formidable research challenge. The chain growth and CO insertion requirements demand multicomponent materials, whose complex reaction dynamics and extensive chemical space defy catalyst design norms. We present an alternative strategy by integrating active learning into experimental workflows, exemplified via the FeCoCuZr catalyst family. Our data-aided framework streamlines navigation of the extensive composition and reaction condition space in 86 experiments, offering >90% reduction in environmental footprint and costs over traditional programs. It identifies the $Fe_{65}Co_{19}Cu_5Zr_{11}$ catalyst with optimized reaction conditions to attain higher alcohol productivities of 1.1 $g_{HA}$ $h^{-1}$ $g_{cat}^{-1}$ under stable operation for 150 h on stream, a 5-fold improvement over typically reported yields. Characterization reveals catalytic properties linked to superior activities despite moderate higher alcohol selectivities. To better reflect catalyst demands, we devise multi-objective optimization to maximize higher alcohol productivity while minimizing undesired $CO_2$ and $CH_4$ selectivities. An intrinsic trade-off between these metrics is uncovered, identifying Pareto-optimal catalysts not readily discernible by human experts. Finally, based on feature-importance analysis, we formulate data-informed guidelines to develop performance-specific FeCoCuZr systems. This approach goes beyond existing HAS catalyst design strategies, is adaptable to broader catalytic transformations, and fosters laboratory sustainability.

Higher alcohol synthesis (HAS) via the thermocatalytic hydrogenation of syngas remains an aspiring technology for defossilizing the production of valuable chemicals and fuel additives while fostering a circular economy and mitigating climate change concerns[1,2]. Although catalyst patents were first filed in the late 1920s[3,4] and a few pilot plants followed sixty years later[5,6]; these efforts have not materialized into a commercial process[5]. Key catalytic performance metrics worth consideration are productivity and selectivity. The highest literature-reported values for space-time yields to higher alcohols ($STY_{HA}$) lie in the vicinity of 0.3 $g_{HA}$ $h^{-1}$ $g_{cat}^{-1}$ [5-7], in contrast to $STY_{HA} > 1$ $g_{HA}$ $h^{-1}$ $g_{cat}^{-1}$ for the related commercialized methanol synthesis, while selectivity to higher alcohols ($S_{HA}$) typically remain below 30%[5,8,9].

The low yields and selectivity of higher alcohols from CO hydrogenation arise from its mechanistic complexity and competing side-reactions which produce appreciable amounts of hydrocarbons and undesired $CH_4$ and $CO_2$[5,6,9]. HAS requires catalytic surfaces with

[1]Institute for Chemical and Bioengineering, Department of Chemistry and Applied Biosciences, ETH Zurich, Vladimir-Prelog-Weg 1, 8093 Zurich, Switzerland. [2]These authors contributed equally: Manu Suvarna, Tangsheng Zou. ✉e-mail: jpr@chem.ethz.ch

**Fig. 1 | Scheme of active learning workflow to develop FeCoCuZr catalysts.** The study is developed in three phases with progressive increases in the number of target performance metrics and variables (right). In each phase, iterative cycles comprising the depicted experimentation and modeling steps are performed (left).

multiple functionalities in close proximity to each other—typically linked to different metals—to simultaneously perform C-O dissociation, C-C coupling, and CO insertion steps over specific active sites[6,9]. Tailored synthesis[10,11], advanced characterization tools such as microscopy[12,13] or operando spectroscopy[14,15], and theoretical modeling[16,17] have been explored to deduce performance descriptors for these complex multimetallic catalysts without achieving the anticipated benefits, necessitating alternative approaches.

The potential of artificial intelligence and machine learning (ML) to handle high-dimensional and intricate systems offers promise to address the above challenge[18–20]. At the intersection of ML and parallel experimentation lies active learning, suitable for expediting material design and process optimization[21–23] through a closed-loop framework of small data−machine intelligence−human decision-making. Although gaining traction in material sciences[24–28], drug discovery and biosystems engineering[29–31], this methodology remains underexplored in catalysis. Notable studies include active learning accelerated density functional theory simulations for screening multimetallics for electrocatalytic $CO_2$ reduction[32] and alkyne semi-hydrogenation[33], based on electronic or structural or descriptors. Experimental studies in optimizing multicomponent catalyst formulations for oxygen reduction[34], methanol synthesis[35], water gas shift[36], and hydrogen evolution[37] reactions have also been reported, but limited to single performance indicator. The predictive nature of these tools offer promise to reduce experimentation time, and more importantly, improve economic and environmental sustainability during catalyst development, which traditionally require hundreds to thousands of screening experiments[35,36,38].

Despite the potential benefits, the applicability of active learning-aided approaches is not yet clear for highly complex catalyst systems for HAS, such as the extensively-investigated modified Fischer-Tropsch (m-FTS) family[2,6]. Comprising Fe[39,40] or Co[12,41] for C-O dissociation and carbon chain growth, in addition to Cu[14,42] which facilitates non-dissociative CO insertion, these active metals have typically been anchored on oxide or carbonaceous supports and further promoted by alkali or noble metals in multicomponent formulations[6,43,44]. Our recent discovery of $ZrO_2$ as a versatile activity promoter for m-FTS systems has led to promising $STY_{HA} > 0.3$ $g_{HA}$ $h^{-1}$ $g_{cat}^{-1}$, with all active

metal pairings (CoCuZr, FeCuZr, FeCoZr) outperforming Zr-free and monometallic counterparts[45]. This prompts further exploration of FeCoCuZr catalysts where interactions between active metals and resultant activity could be further enhanced. However, the extensive chemical space and complex reaction dynamics of these materials, and the lack of established descriptors, pose a significant challenge.

In this study, we pioneer an active learning strategy to accelerate the development of highly active FeCoCuZr catalysts. Key features include (i) high prediction power leading to the $Fe_{65}Co_{19}Cu_5Zr_{11}$ catalyst and its optimal reaction conditions with stable $STY_{HA}$ of 1.1 $g_{HA}$ $h^{-1}$ $g_{cat}^{-1}$ for at least 150 h, the highest reported for direct HAS from syngas, (ii) a substantial reduction in time and resources by identifying the optimal system in 86 experiments from the vast space of ca. five billion potential combinations, (iii) multi-objective optimization to reveal intrinsic performance trade-offs and recommended Pareto-optimal catalysts that minimize selectivity towards $CO_2$ and $CH_4$ while still maintaining high $STY_{HA}$. These results underscore the potential of data-driven approaches to sustainably expedite efficient multi-component catalyst development and foster innovation in catalysis research[32,36].

## Results

### Overview and scope of the active learning framework

We devised an active learning approach by integrating data-driven algorithms with experimental workflows, which continuously learns from existing and newly-generated data from iterative experimental cycles, to explore and identify FeCoCuZr compositions and reaction conditions optimizing catalyst performance metric(s) of interest (Fig. 1)[36,37,46]. The core of the data-driven model combines Gaussian process (GP) and Bayesian optimization (BO) algorithms, along with human decision-making in order to accomplish single or multi-objective tasks[46,47].

To showcase the feasibility of this approach to HAS, the study was systematically conducted in three distinct phases by progressively increasing the model complexity. In Phase 1, the catalyst composition was varied with the objective of maximizing $STY_{HA}$ at fixed reaction conditions. In Phase 2, the dimensionality of the problem was increased by concurrently exploring the catalyst compositions and

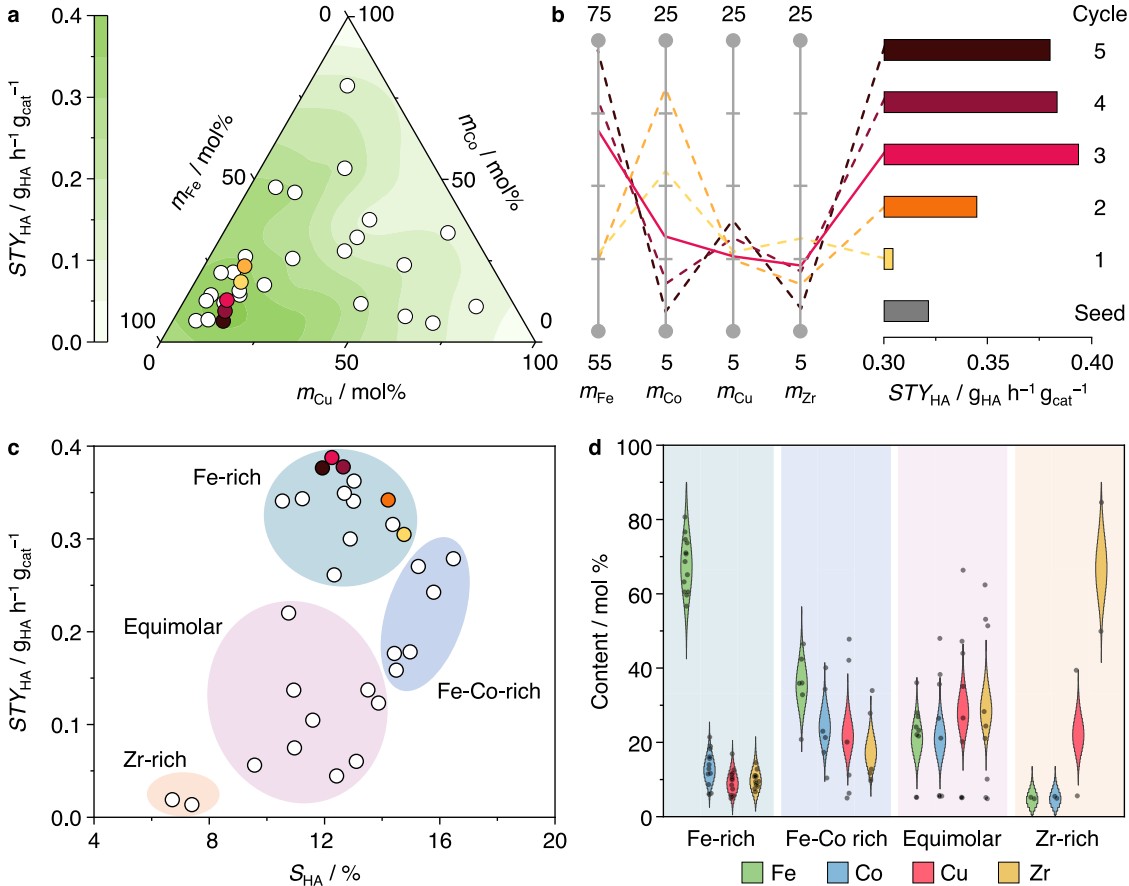

**Fig. 2 | Exploration of catalyst compositions to maximize $STY_{HA}$ at fixed reaction conditions. a** Performance plot based on the chemical space of FeCoCuZr catalysts, with the sum of Fe, Co, and Cu molar contents normalized to 100% and excluding the Zr content. Compositions of catalysts evaluated during five active learning cycles are depicted as circles, and contours are generated from measured $STY_{HA}$ values of selected catalysts containing $10 \pm 5$ mol% Zr. **b** Evolution of the performance obtained from the best catalyst in each cycle and its associated composition (units in mol%). The catalysts are benchmarked vs. the reference $Fe_{79}Co_{10}Zr_{11}$ catalyst from seed experiments. **c** Performance-based clustering of the catalysts. The catalysts with the highest $STY_{HA}$ in each cycle are colored in **a**–**c**. **d** Violin plots of catalyst compositions for the clusters identified in **c**, with individual data points shown as gray dots. Source data are provided in the source data file.

reaction conditions to maximize $STY_{HA}$. This approach was subsequently extended towards multi-objective capabilities by simultaneously maximizing $STY_{HA}$ while minimizing combined selectivity to carbon dioxide and methane ($S_{CO_2 + CH_4}$) in Phase 3. Iterative cycles comprising of six experiments were conducted during each phase, until the target performance metric(s) were achieved or reached saturation.

### Phase 1: Optimal catalyst formulations for productivity

The first phase aimed to explore the suitability of the active learning framework in identifying optimal FeCoCuZr formulations for maximizing $STY_{HA}$ under fixed reaction conditions, specifically the $H_2$:CO ratio ($H_2$:CO), reaction temperature ($T$), pressure ($P$), and the gas hourly space velocity ($GHSV$). This strategy enabled the exploration of a space containing >175,000 unique compositional possibilities, known as the chemical space[34,36] (Supplementary Note 1), and helped understand the sensitivity of higher alcohols productivity to catalyst composition in this family of materials. Without a priori composition and performance data for FeCoCuZr formulations, the initial model training was performed using 31 data points on the FeCoZr, FeCuZr, and CuCoZr catalysts recently reported by our group[45], denoted as seed experiments for Phase 1 (Supplementary Note 2, Supplementary Table 1). Reaction conditions were fixed at $H_2$:CO = 2.0, $T$ = 533 K,

$P$ = 50 bar, and $GHSV$ = 24,000 cm$^3$ h$^{-1}$ g$_{cat}^{-1}$ across cycles to match those in the seed dataset (Supplementary Note 2).

In each cycle, the GP-BO model was trained using the molar content values of the four elements (Fe, Co, Cu, Zr) and the corresponding $STY_{HA}$ of all catalysts in the dataset. Subsequently, we evaluated the expected improvement (EI) and predictive variance (PV) acquisition functions separately under specific constraints to generate candidate compositions (see Methods section, "Gaussian process and Bayesian optimization", Supplementary Table 2). Six suitable catalysts were manually selected for experimentation by balancing the number of recommendations from EI, which searches for compositions maximizing the $STY_{HA}$ objective (i.e., exploitation), and from PV, which seeks potential candidates in the unexplored chemical space (i.e., exploration)[23,48]. Here, it is important to acknowledge the key role of human decision-making in providing a judgement and selection from the suggested compositions, which allowed us to supervise and fine-tune the implementation of active learning at this early stage. The experimentally evaluated performance together with measured compositions of these six catalysts were added to the dataset to re-train the model for the next cycle.

Five iterative cycles were performed (30 catalysts, see Supplementary Tables 3-7), which mapped the chemical space and identified regions with high $STY_{HA}$ (Fig. 2a). Progressive improvements in $STY_{HA}$ across cycles were achieved (Fig. 2b, Supplementary Tables 3-7), with

the $Fe_{69}Co_{12}Cu_{10}Zr_9$ catalyst in Cycle 3 attaining the highest $STY_{HA} = 0.39$ $g_{HA}$ $h^{-1}$ $g_{cat}^{-1}$, a 1.2-fold improvement over the $Fe_{79}Co_{10}Zr_{11}$ seed benchmark ($STY_{HA} = 0.32$ $g_{HA}$ $h^{-1}$ $g_{cat}^{-1}$). This performance was retained for at least 100 h on stream with no visible sign of deactivation (Supplementary Fig. 1). Similar compositions and performances obtained by the best catalysts in Cycles 4 and 5 confirmed the convergence of results.

To reveal compositional trends driving performance, we used the $k$-means clustering algorithm[21,34]. This allowed us to identify catalysts that had high $STY_{HA}$ and $S_{HA}$ - both being key performance metrics, and enabled informed decision making regarding experiment selection in subsequent phases. Four distinct clusters were observed (Fig. 2c, d). Catalysts in the Zr-rich and equimolar clusters exhibit low $STY_{HA}$, likely due to low contents and suboptimal ratios of the active metals, respectively. Fe stands out as a key active metal, as maximum $STY_{HA}$ up to 0.39 $g_{HA}$ $h^{-1}$ $g_{cat}^{-1}$ was attained by Fe-rich catalysts, while those in the Fe-Co rich cluster exhibit the highest $S_{HA}$ up to 17%. Notably, Zr content converged towards 10% in the highest performing Fe-rich catalysts, mimicking that obtained over the seed catalysts[45]. Irrespective of compositional differences, product distributions of best performing catalysts in each cycle largely resembled each other, with $S_{HA} = 14 \pm 2\%$, $S_{CO_2} = 13 \pm 5\%$, $S_{CH_4} = 25 \pm 5\%$, and $(S_{CH-} + S_{CH=}) = 48 \pm 4\%$ (Supplementary Fig. 2, Supplementary Table 8).

The superior $STY_{HA}$ of the $Fe_{69}Co_{12}Cu_{10}Zr_9$ optimal catalyst compared to the $Fe_{79}Co_{10}Zr_{11}$ seed benchmark stemmed from an increase in $X_{CO}$ (45% vs. 36%), since they showed similar $S_{HA} \sim 13\%$. We thus conducted characterization studies to identify key features behind the activity gain. STEM-EDX maps of both catalysts (Fig. 3a, b, Supplementary Figs. 3, 4) identified small domains of $ZrO_2$ in intimate contact with larger active metal nanoparticles. These defect-rich $ZrO_2$ domains, as indicated by the presence of $Zr^{\delta+}$ contributions in the Zr 3$d$ region of XPS spectra (Fig. 3c), were previously identified to enhance surface iron and cobalt carbide formation in line with the C 1$s$ signal of XPS spectra (Fig. 3d)[49]. Carbides are known active phases in CO hydrogenation, and their interface with partially reduced iron oxide species is thought to enhance non-dissociative CO activation and higher alcohol formation in iron-rich catalysts[45,49,50]. Notably, elemental distributions of Fe and Co suggest that these two metals are inter-dispersed and not segregated in both catalysts whether as calcined or after use (Fig. 3a, b, Supplementary Figs. 3, 4), in line with their tendency to form intermetallic alloys or mixed oxides. Initially dispersed Cu in the fresh $Fe_{69}Co_{12}Cu_{10}Zr_9$ catalyst agglomerated into distinct 10–20 nm-sized agglomerates in contact with Fe-Co nanoparticles following reduction and reaction (Fig. 3b, Supplementary Figs. 3, 4), accompanied by a decrease in surface area (Supplementary Table 9) and confirmed by XRD patterns (Supplementary Fig. 5). Such architectural features have also been observed for similar FeCoCu materials prepared by a sol-gel method without Zr present in the composition[51].

The presence of copper enhances the surface reducibility of $Fe_{69}Co_{12}Cu_{10}Zr_9$ as demonstrated by $H_2$-TPR (Fig. 3e), where oxidic Cu is readily reduced in the presence of $H_2$ to the metallic state, which in turn enhances hydrogen splitting and spillover to neighboring Fe-Co oxide domains[51]. Temperature-programmed $H_2$-$D_2$ exchange experiments also showed a 40 K decrease in the exchange temperature for $Fe_{69}Co_{12}Cu_{10}Zr_9$ (Fig. 3f, g) compared to $Fe_{79}Co_{10}Zr_{11}$, suggesting that Cu nanoparticles improve hydrogen activation and thus play a similar role as in other Cu-catalyzed reactions such as methanol or olefin synthesis from $CO_x$[51,52]. As such, the increase in $X_{CO}$, formation rates of each alcohol (Supplementary Table 10) and therefore $STY_{HA}$ attainable over iron-rich FeCoCuZr catalysts originates from its enhanced $H_2$ activation ability from copper, while retaining the characteristics imparted by well-mixed, carbide-rich Fe-Co phases promoted by dispersed and defective $ZrO_2$. Under reaction conditions, $*CH_x$, $CH_3O*$, and $CH_3CH_2O*$ intermediates were detected by in situ DRIFTS

(Supplementary Fig. 6), in line with the expected mechanisms of $*CH_x$ coupling and non-dissociative $*CO$ insertion for HAS over m-FTS catalysts.

It is worth noting that the model formulation did not include any explicit chemical information guiding the iterative cycle. Nevertheless, it was able to provide performance predictions with high accuracy. The overall performance was influenced by the balance between EI and PV functions used in the GP-BO algorithm[23,24,48], as the latter exhibits higher uncertainty leading to lower accuracy, and vice versa for the former. A total of 13 and 17 catalysts were evaluated based on recommendations from the EI and PV acquisition functions, respectively (Supplementary Fig. 7, Supplementary Table 11), progressively improving model performance from an initial coefficient of determination $R^2 = 0.36$ in Cycle 1 to $R^2 = 0.84$ by the final cycle. This improvement resulted from the expansion of available data generated during the active learning cycles and an increased number of experimental evaluations guided by the exploitation function probing regions of high performance. Accurate predictions on catalyst performance could be made owing to the standardized synthesis method that ensured consistency in structural properties.

## Phase 2: Optimal catalyst formulations and reaction conditions for productivity

While basic reactivity patterns and the relevance of operating conditions are well known for HAS, there are no universally applicable set of optimal conditions as the influence of each parameter is catalyst-specific. The second phase of this study tackles this by expanding the exploration space to include reaction conditions, including $H_2$:CO, $T$, and $GHSV$, defined as the parametric space (Supplementary Note 1). As Phase 1 did not include variation of reaction conditions, 20 additional experiments were performed to broaden the range of reaction conditions initially covered by the model, denoted as seed experiments for Phase 2 (Supplementary Note 3, Supplementary Table 12). Phase 2 was initiated by training the GP-BO algorithm using these 50 data points under compositional and parametric constraints based on knowledge from previous experiments and the literature (Supplementary Note 4, Supplementary Table 13).

Reaction conditions were found to exert a significant impact on $STY_{HA}$, as $Fe_{61}Co_{20}Cu_9Zr_{10}$ identified as the best performer in Cycle 1 was already able to exceed 0.5 $g_{HA}$ $h^{-1}$ $g_{cat}^{-1}$ at $H_2$:CO = 1.8, $T = 552$ K, $P = 50$ bar, and $GHSV = 32,550$ $cm^3$ $h^{-1}$ $g_{cat}^{-1}$, almost 1.5-fold higher compared to the seed data used in Phase 1. Over the next two active learning cycles, $STY_{HA}$ reached ~0.7 $g_{HA}$ $h^{-1}$ $g_{cat}^{-1}$, nearly doubling productivity compared to the maximum achieved in Phase 1 (Fig. 4a, Supplementary Tables 14–16). We noticed that by the end of Cycle 3, the optimizer was locally constrained at the $GHSV$ upper bound of 50,000 $cm^3$ $h^{-1}$ $g_{cat}^{-1}$. As it is typically observed in literature that an increase in $GHSV$ concurrently increases $STY_{HA}$ owing to higher reactant flows despite a slight reduction in $X_{CO}$[5,7,9], the upper bound was set to 100,000 $cm^3$ $h^{-1}$ $g_{cat}^{-1}$ in Cycle 4 to observe the behavior of the model. This adjustment led to the GP-BO extrapolating to previously unexplored $GHSV$ values, suggesting catalytic systems that provided higher $STY_{HA}$ of up to 0.9 $g_{HA}$ $h^{-1}$ $g_{cat}^{-1}$. By Cycle 5, our framework recommended the highly active $Fe_{65}Co_{19}Cu_5Zr_{11}$ catalyst that attained $STY_{HA} = 1.1$ $g_{HA}$ $h^{-1}$ $g_{cat}^{-1}$ at operating conditions of $H_2$:CO = 2.2, $T = 551$ K, $P = 50$ bar, and $GHSV = 90,000$ $cm^3$ $h^{-1}$ $g_{cat}^{-1}$, marking a significant 3.5-fold increase from the original Phase 1 seed benchmark (Fig. 4a, Supplementary Table 18). The stability of this catalyst was evaluated in a 150-h catalytic run (Fig. 4b), where $X_{CO} \geq 40\%$ and $STY_{HA} \geq 1$ $g_{HA}$ $h^{-1}$ $g_{cat}^{-1}$ were maintained throughout. Phase 2 concluded upon the completion of Cycle 6, during which catalytic systems yielding approximately 1 $g_{HA}$ $h^{-1}$ $g_{cat}^{-1}$ were achieved once more, suggesting repeatability of results as well as model saturation (Fig. 4a, Supplementary Table 19).

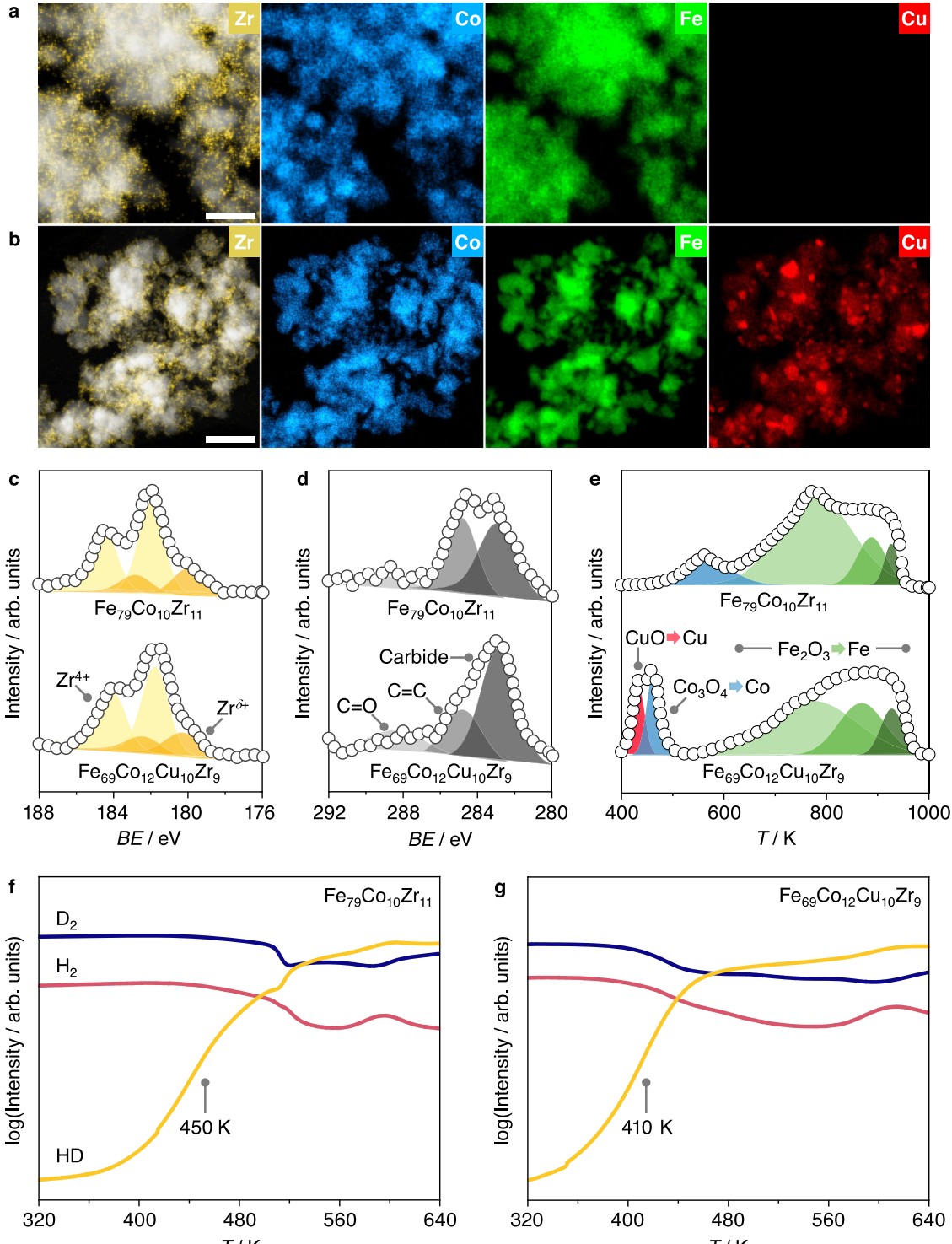

**Fig. 3 | Characterization of the best performing catalyst in Phase 1 and the reference seed catalyst.** STEM-EDX elemental maps after reaction of (**a**) the reference $Fe_{79}Co_{10}Zr_{11}$ and (**b**) the best performing $Fe_{69}Co_{12}Cu_{10}Zr_9$ catalyst identified in Phase 1. Scale bars represent 50 nm. XPS spectra around (**c**) Zr 3*d* and (**d**) C 1*s* regions after reaction with deconvoluting signals indicated. **e** $H_2$-TPR profiles with deconvoluted peaks corresponding to the reduction of oxides indicated. For iron oxide: $Fe_2O_3 \rightarrow Fe_3O_4 \rightarrow FeO \rightarrow Fe$. Temperature-programmed $H_2$-$D_2$ exchange profiles of (**f**) $Fe_{79}Co_{10}Zr_{11}$, (**g**) $Fe_{69}Co_{12}Cu_{10}Zr_9$ with the exchange temperature (at the maximum rate of HD formation) marked. Reaction conditions: $H_2$:CO = 2.0, $T = 533$ K, $P = 50$ bar, and $GHSV = 24,000$ cm³ h⁻¹ g$_{cat}$⁻¹. Source data are provided in the source data file.

Benchmarking FeCoCuZr catalytic systems from Phases 1 and 2 with literature-reported catalysts across various families such as Rh-based, Mo-based, and m-FTS-based revealed notable differences. The literature-reported catalysts (a total of 125 catalysts were examined) exhibited an average $STY_{HA} \approx 0.1$ g$_{HA}$ h⁻¹ g$_{cat}$⁻¹ with top performers in the 90[th] percentile reaching 0.18 g$_{HA}$ h⁻¹ g$_{cat}$⁻¹. Conversely, the best performing FeCoCuZr catalysts in the different cycles of Phases 1 and 2 had an average $STY_{HA}$ of approximately 0.6 g$_{HA}$ h⁻¹ g$_{cat}$⁻¹ highlighting significantly enhanced productivity compared to literature-reported counterparts for direct hydrocarbon synthesis from syngas for direct

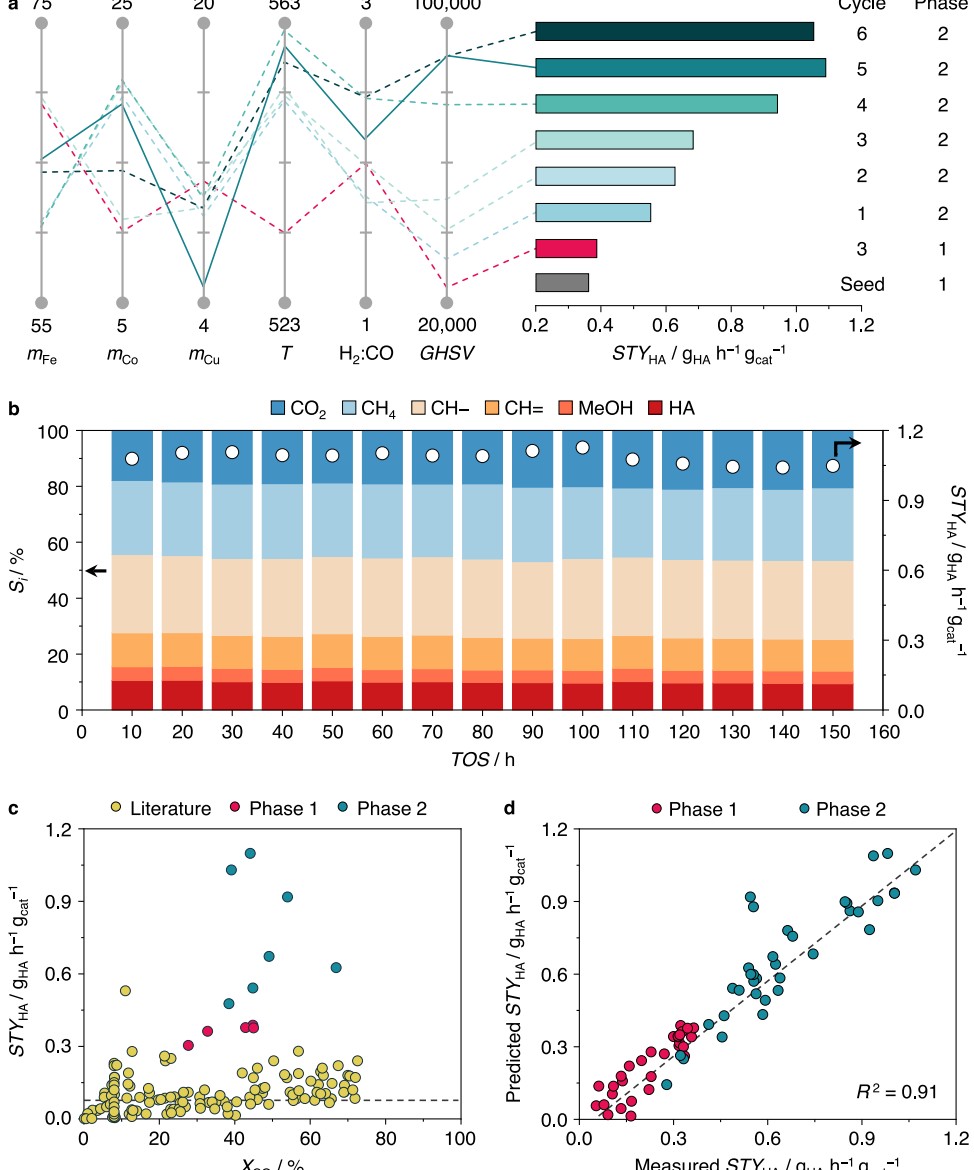

**Fig. 4 | Identification of catalyst compositions and reaction conditions maximizing $STY_{HA}$. a** Evolution of the performance obtained from the best catalyst in each cycle and its respective composition and reaction conditions. All reactions were conducted at $P = 50$ bar. **b** Stability test of the most productive $Fe_{65}Co_{19}Cu_5Zr_{11}$ catalyst over 150 h on stream. **c** Comparison of the performance of best FeCoCuZr catalytic systems developed in each active learning cycle across Phase 1 and 2, with over 120 Rh-, Mo-, and m-FTS based catalysts reported in literature. The horizontal dashed line represents average literature reported $STY_{HA}$ values. **d** Parity plots depicting the model performance through the active learning phases. Source data are provided in the source data file.

HAS from syngas[6,8] (Fig. 4c). For comparative analysis, selectivity to higher alcohols exhibited a less responsive nature than productivity in this family. Regardless of the diverse set of compositions and reaction conditions investigated, $S_{HA} = 11 \pm 2\%$ was the mean of the most active systems in Phase 2 (Supplementary Fig. 8, Supplementary Table 20), similar to the observation made in Phase 1 and pointing to an intrinsic feature of the FeCoCuZr family.

The impact of varying reaction conditions on performance in Phase 2 could be visualized by mapping catalyst compositions and their respective $STY_{HA}$ attained in four clusters dictated by $GHSV$ and $T$ ranges (Supplementary Fig. 9). The highest-performing catalytic systems share similar compositions to the iron-rich catalysts discovered to be optimal in Phase 1 (Fig. 2c, d), hinting that they retain the same catalytic features previously determined to boost activity. The model recommendations of maximizing $GHSV$, $T = 550$–$570$ K, and moderate $H_2$:CO ~ 2 are also in line with established heuristics for syngas-based

HAS (Supplementary Note 4). The accuracy remained high with $R^2 = 0.78$ in Phase 2, comparable to that in Phase 1. Progressive improvements in accuracy were evident in Phase 2, with the mean absolute percentage error (MAPE) between predicted and measured $STY_{HA}$ for each cycle in Phase 2 decreasing from 33% in Cycle 1 to 7.6% in Cycle 6 (Supplementary Table 21). Considering predicted and measured $STY_{HA}$ across both phases resulted in an overall performance accuracy of $R^2 = 0.91$ (Fig. 4d) with a low root mean squared error (RMSE) of 0.09 g h$^{-1}$ g$_{cat}$$^{-1}$.

## Phase 3: Maximized productivity and minimized selectivity to by-products

The third phase of this study aimed to apply active learning to search for catalytic systems that could meet multiple performance criteria, better reflecting the real-world demands on catalysts. Given the modest $S_{HA}$ across all catalysts developed in Phase 1 and 2, we focused

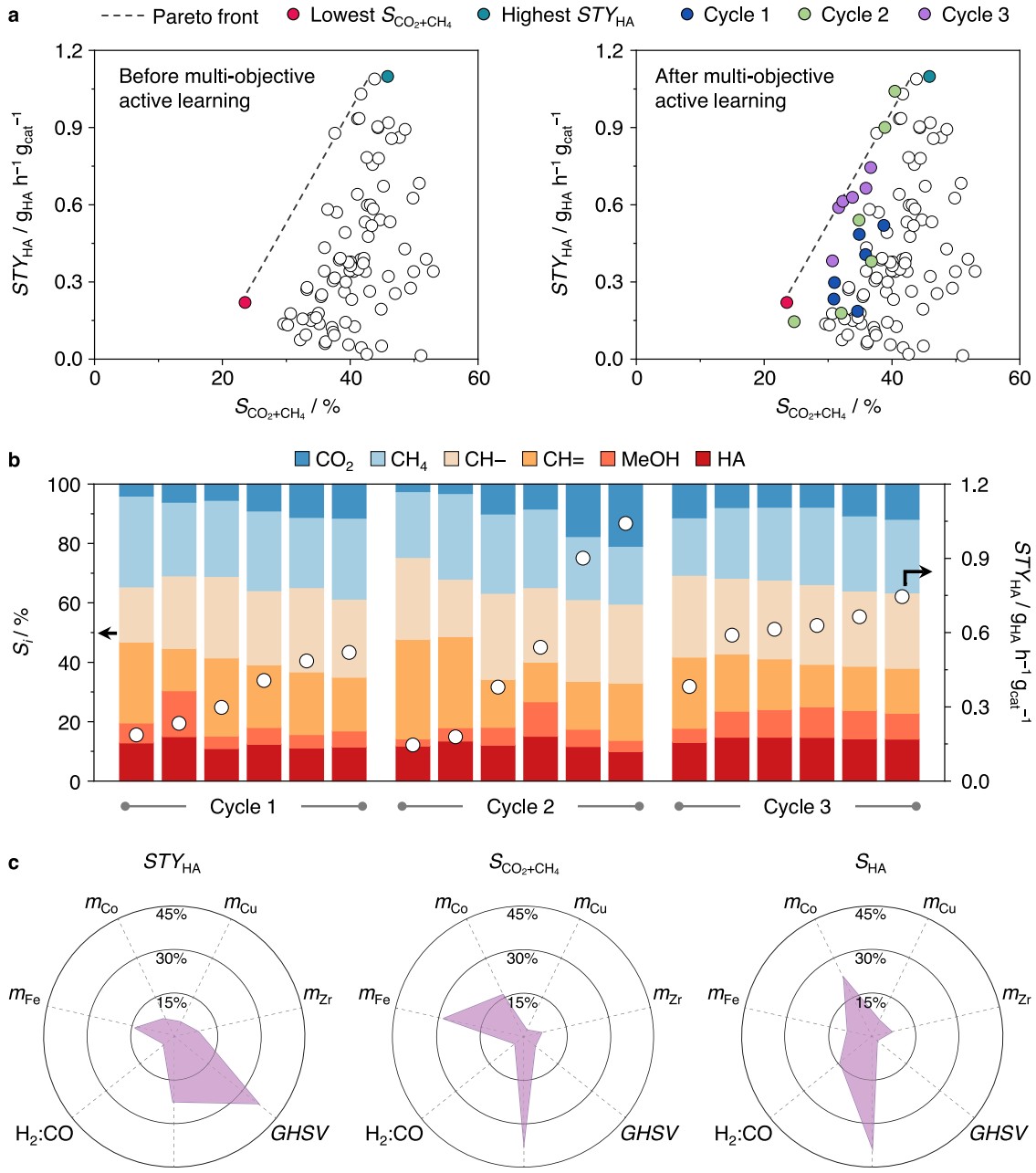

**Fig. 5 | Uncovering Pareto-optimal catalysts and performance drivers. a** Visual depiction of Pareto-optimal catalytic performances before and after multi-objective active learning. The Pareto front delimits the zone available by catalysts developed in Phase 1 and 2. **b** Product distribution for FeCoCuZr catalysts across Phase 3. **c** Relative feature-importance of catalyst compositions and reaction conditions on each of the performance metrics as determined by SHAP analysis. Source data are provided in the source data file.

on selectivities towards carbon dioxide ($S_{CO_2}$) and methane ($S_{CH_4}$), considered as the least valuable products in HAS[5,9]. The 86 data points in Phase 1 and 2 exhibited $S_{CO_2} = 16 \pm 6\%$ and $S_{CH_4} = 25 \pm 4\%$, highlighting the significance of the water-gas shift (WGS) and CO methanation reactions, especially at conditions favoring high $X_{CO}$ and therefore $STY_{HA}$. (Supplementary Note 4, Supplementary Fig. 10). A plot of $STY_{HA}$ vs. $S_{CO_2 + CH_4}$ suggested an intrinsic trade-off in the form of a Pareto front (Fig. 5a), in which the improvement of one metric would likely be at the expense of the other[53,54].

This scenario was explored in Phase 3 by varying the catalyst compositions and reaction conditions simultaneously to maximize $STY_{HA}$ and minimize $S_{CO_2 + CH_4}$. For this purpose, the GP-BO algorithm was trained with data from Phases 1 and 2 with $STY_{HA}$ and $S_{CO_2 + CH_4}$ as

target metrics, using the Expected Hypervolume Improvement (EHVI) acquisition function, which guides the optimization process to recommend catalyst composition and reaction conditions that are likely to lead to better trade-offs among conflicting objectives[23,35].

During Cycle 1, a significant discrepancy between predicted and measured values of $STY_{HA}$ and $S_{CO_2 + CH_4}$ was observed (Supplementary Fig. 11, Supplementary Table 26) with none of the catalysts evaluated near the Pareto barrier (Fig. 5a, Supplementary Table 22). Upon entering Cycle 2, two of the six catalysts evaluated were situated on the Pareto front (Fig. 5a, Supplementary Table 23), while the model exhibited enhanced prediction accuracy. The highest performing system notably attained $STY_{HA} = 1.04$ $g_{HA}$ $h^{-1}$ $g_{cat}^{-1}$, only 5% lower than the maximum attained in Phase 2 but with a drastically reduced

$S_{CO_2+CH_4} = 40\%$ vs. 46%. By Cycle 3, model recommendations improved significantly as five of the six catalysts evaluated lie directly on the Pareto frontier without crossing it (Fig. 5a, Supplementary Table 24) attaining $S_{CO_2+CH_4} = 34 \pm 2\%$ and $STY_{HA} = 0.65 \pm 0.05$ $g_{HA}$ $h^{-1}$ $g_{cat}^{-1}$. Herein, while the productivity remained ca. two times higher than average literature values, we highlight the selectivity of undesired $CO_2$ and $CH_4$ was minimized by around 10% (Fig. 5b), in comparison to some of the catalysts developed in Phase 2, suggesting an optimal trade-off between $STY_{HA}$ and $S_{CO_2+CH_4}$. Notably, we identified optimal systems along the Pareto frontier, suggesting an intrinsic limitation of this family of HAS catalysts to achieve $S_{CO_2+CH_4} < 30\%$ without compromising $STY_{HA}$. However, within this constraint, our strategy eventually uncovered five Pareto-optimal catalytic systems which are otherwise non-intuitive and not easily accessed by human experts[55], thereby underscoring its versatility and significance.

## Performance drivers and data-informed guidelines

We sought to elucidate the main performance drivers among the set of input features impacting $STY_{HA}$ and $S_{CO_2+CH_4}$. However, an inherent challenge with most ML algorithms, including the GP regressor used in this study, lies in the complexity of deciphering the internal rationale behind predictions—rendering them black-box in nature. To address this challenge and make the model interpretable, we utilized the agnostic ML explainer, SHapley Additive exPlanations (SHAP)[56]. This methodology facilitates the extraction of interpretable insights from the GP algorithm through the computation of feature-importance scores[57,58]. Akin to sensitivity analysis, SHAP determines the individual or combined contributions of features to the model's prediction, enabling catalysis practitioners to quantify the relative importance of different features affecting performance, that can be corroborated with existing knowledge or lead to testable hypotheses.

The overall influence of each feature was expressed by normalized SHAP values, revealing that reaction conditions and catalyst compositions contributed to ca. 60% and 40%, respectively, to the model predictions for both targeted metrics in Phases 2 and 3 (Fig. 5c). In the case of $STY_{HA}$, $GHSV$ and $T$ emerged as the two most important parameters, in alignment with earlier intuition in Phase 2 as well as findings in the literature for similar $C_1$ transformations[36,57]. Fe content ranked as the most prominent compositional input, in line with the high productivities only attained by iron-rich catalysts, as highlighted in the discussion of Phase 1 and given the claimed role of Fe phases in C-C coupling[6]. For $S_{CO_2+CH_4}$, $T$ was identified as the most significant variable, followed by Fe and Co contents. The role of $T$ in dictating selectivity patterns could be ascribed to the coexistence of competing reaction networks, each with different temperature-dependent kinetic and thermodynamic barriers including HAS, WGS, or methanation. This highlights the importance of optimizing $T$ to fine-tune the selectivity towards higher alcohols or by-products. The Fe-Co-rich cluster determined in Phase 1 (Fig. 2c) catered most favorably to HA selectivity, with Fe-Co surface carbides previously identified as a key feature for selective higher alcohol production from catalyst characterization[45].

Despite demonstrating the efficacy of active learning in uncovering catalytic systems that enhance multiple performance metrics for HAS, it is essential to acknowledge its scope and limitations in its current form. The lack of electronic or structural descriptors as inputs to the model and its inability to optimize performance metrics which are intrinsically unresponsive to screened variables, such as $S_{HA}$ in this work, can be mentioned (see Supplementary Note 5 for extended discussion). Nonetheless, in the course of this study, three categories of catalytic systems within the FeCoCuZr family emerged exhibiting distinct performance characteristics, namely high $STY_{HA}$ and $S_{CO_2+CH_4}$ ($STY_{HA} = 0.97 \pm 0.08$ $g_{HA}$ $h^{-1}$ $g_{cat}^{-1}$, $S_{CO_2+CH_4} = 44 \pm 2\%$, $S_{HA} = 10 \pm 1\%$), low $STY_{HA}$ and $S_{CO_2+CH_4}$ ($STY_{HA} = 0.25 \pm 0.07$ $g_{HA}$ $h^{-1}$ $g_{cat}^{-1}$, $S_{CO_2+CH_4} = 31 \pm 3\%$, $S_{HA} = 14 \pm 2\%$), and Pareto-optimal catalysts

($STY_{HA} = 0.63 \pm 0.06$ $g_{HA}$ $h^{-1}$ $g_{cat}^{-1}$, $S_{CO_2+CH_4} = 34 \pm 2\%$, $S_{HA} = 14.6 \pm 0.3\%$) (Fig. 6). Each category favors unique catalyst compositions and reaction conditions; for instance, systems displaying high $STY_{HA}$ and $S_{CO_2+CH_4}$ are characterized by high molar Fe content, $H_2$:CO, and $GHSV$ values, whereas the low $STY_{HA}$ and $S_{CO_2+CH_4}$ counterparts are favored at equimolar Fe-Co contents, low $H_2$:CO, and milder $T$. The Pareto-optimal catalysts feature a combination of the aforementioned traits, recommending high Fe contents and operation at high $GHSV$ and mild $T$. These quantitative guidelines, especially those relating to operating conditions, align with literature findings and are likely not dependent on specific catalyst formulations and could be relevant to HAS catalysts in general[6,9]. However, the exact compositional guidelines provided herein apply to the FeCoCuZr catalysts investigated in this study and would arguably not be directly relevant in designing HAS systems with different active metals, promoters, and architectures prepared by different synthesis methods. Importantly, this methodology based on data analysis can be extended to other potential HAS systems or even other multi-product chemical transformations, provided sufficient experimental data is available. Other users are thus recommended to formulate specific guidelines for their application during the active learning process. Overall, in the absence of quantifiable techno-economic data and community consensus on practically relevant productivities or selectivities for HAS, this approach provides guidelines for optimizing key metrics, serving as valuable assets to catalysis practitioners and industry stakeholders to accelerate research efforts by assisting in the selection of appropriate catalytic systems and experiments, ultimately saving time and resources.

## Active learning and sustainable laboratories

While the possible chemical and parametric space of the FeCoCuZr systems is in the order of billion combinations, practical and real-world studies on multicomponent catalysts range between hundreds to thousands screening experiments[35,36,38]. By employing active learning we mapped the vast space of FeCoCuZr catalysts to a cumulative 104 experiments, across Phases 1–3 to meet the desired performance objectives, confirming the growing body of literature that claim active learning accelerates experimental efforts[34–36]. This has a profound impact on the environmental and economic sustainability of catalyst development programs that has not been explored.

To this context, assuming this study as representative of a catalyst development endeavor, we assessed the degree to which active learning could impact both sustainability pillars in laboratories (see scope in Supplementary Note 6). Our analysis suggests average reductions exceeding 90% in carbon footprint and costs on benchmarking with traditional campaigns (Fig. 7, Supplementary Tables 27–29). We also observe a very mild dependency of this result with regional variations across the globe affecting, for example, composition of the energy mix or laboratory operational expenditure (Supplementary Fig. 12). Thus, by reducing consumption of chemicals and energy, and optimizing resource utilization, active learning remarkably fosters sustainable catalysis laboratories.

## Discussion

In this study, we advance research on multimetallic catalysts for HAS, exemplified by the novel FeCoCuZr catalyst family, by integrating digital tools in experimental catalysis in the form of active learning. It drastically narrows down the vast chemical and reaction condition space of multimetallic systems, in our case from around $5 \times 10^9$ combinations to 86 experiments, significantly accelerating experimental workflows. This strategy identified the $Fe_{65}Co_{19}Cu_5Zr_{11}$ catalyst and its optimized reaction conditions ($H_2$:CO = 2.2, $T$ = 551 K, $P$ = 50 bar, and $GHSV$ = 90,000 $cm^3$ $h^{-1}$ $g_{cat}^{-1}$) able to achieve $STY_{HA} = 1.1$ $g_{HA}$ $h^{-1}$ $g_{cat}^{-1}$, the highest reported in literature for direct thermocatalytic HAS from syngas.

**Fig. 6 | Establishment of guidelines for developing performance-specific catalysts.** Data-informed guidelines aimed towards the development of FeCoCuZr catalytic systems belonging to three performance categories. The box plots present a statistical summary of the composition and reaction condition requirements for each category, showing the minimum, interquartile range, and maximum values. Outliers are indicated as points outside the boundary of the whiskers wherever applicable. The corresponding performance metrics are shown below as horizontal bars. Source data are provided in the source data file.

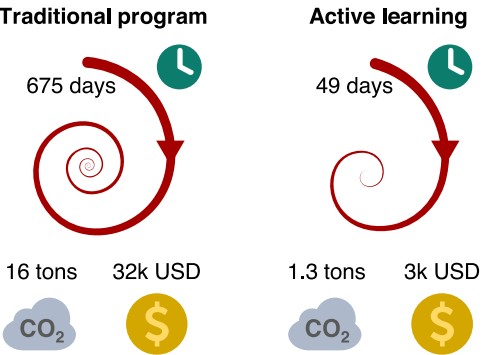

**Fig. 7 | Impact of active learning on laboratory sustainability.** Schematic illustration of the reduction in work-days, $CO_2$ footprint, and operating costs on a global average basis fostered by the adoption of active learning framework over traditional catalyst development programs, calculated from results obtained in this study. The spirals represent experimental efforts, and the derivation of the sustainability metrics shown are detailed in Supplementary Note 6 and regional variations in Supplementary Fig. 12.

Representative catalytic materials were characterized to uncover reasons behind the outstanding performance, discovering the interplay between $H_2$ activation on Cu and well-mixed, carbide-rich Fe-Co nanoparticles to boost activities. This strategy was also successful in identifying catalysts optimizing multiple performance metrics, by adapting the model to search for systems maximizing $STY_{HA}$ while minimizing $S_{CO_2 + CH_4}$, in view of the insensitivity of $S_{HA}$ to compositional and operational variations for this family of materials. This task uncovered an intrinsic trade-off between the two performance metrics and aided in the discovery of Pareto-optimal catalysts, a challenge not easily tackled by human experts. Finally, key performance drivers were quantified via SHAP analysis, and based on the insights gathered, formulate data-informed guidelines to assist catalysis practitioners in developing performance-specific FeCoCuZr catalysts for HAS.

The presented active learning framework can be expanded to carry out further optimization within this catalyst family such as adding carrier materials or promoters. Nevertheless, given the data-driven nature of this active learning framework, independent of the reaction or specific catalyst properties, it can be extended beyond the FeCoCuZr family and HAS. From a broader perspective, this work demonstrates active learning as an innovative tool that can go beyond traditional design strategies

to largely accelerate catalyst development, while minimizing economic and environmental resource utilization, advocating towards more sustainable laboratories.

## Methods

### Catalyst synthesis

$Fe_xCo_yCu_zZr_a$ catalysts with molar Fe, Co, Cu, and Zr contents of $x$, $y$, $z$ and $a$ (metals basis, units of mol%), respectively, were prepared via a sol-gel method. Appropriate amounts of metal precursors [$Fe(NO_3)_3 \cdot 9H_2O$ (Sigma-Aldrich, 98%), $Co(NO_3)_2 \cdot 6H_2O$ (abcr, 98%), $Cu(NO_3)_2 \cdot 3H_2O$ (Sigma-Aldrich, 98%), and $ZrO(NO_3)_2 \cdot xH_2O$ (Thermo Fischer Scientific, 99.5%)] required for a total metal concentration of 0.25 M with the targeted nominal metal contents were dissolved in 100 $cm^3$ of ethanol (Fisher Scientific, 99.8%) in a 500 $cm^3$ round bottom flask. A 100 $cm^3$ solution of 0.5 M L-(+)-tartaric acid (99.5%, Sigma-Aldrich) was added dropwise to the metal precursor solution under magnetic stirring (ca. 1000 rpm) for 12 h. The resulting gel containing a metal-tartrate complex was separated from excess solvent using a Büchi Rotavapor R-114 instrument at 323 K and 100 mbar. The colloids obtained were dried in a vacuum oven at 383 K for 12 h, ground into a fine powder and calcined in static air at 673 K (2 K min$^{-1}$) for 3 h.

### Catalyst evaluation

The direct conversion of syngas to higher alcohols was conducted using a PID Eng&Tech high-pressure continuous-flow setup comprising four parallel fixed-bed reactors (main features in Supplementary Fig. 13). Sieved powder catalyst as calcined (50 mg, particle size = 0.15−0.30 mm) were diluted with quartz sand (75 mg, particle size = 0.15−0.30 mm) and loaded into each reactor tube (internal diameter 4 mm), held in place by a 2 mm bed of quartz wool set on a quartz frit. Catalysts were activated in situ to reduce surface metal oxides under a 20 $cm^3$ min$^{-1}$ flow of 50 vol.% $H_2$ (PanGas, purity 5.0) in Ar (PanGas, 5.0) at atmospheric pressure and $T = 623$ K (5 K min$^{-1}$) for 4 h. After the reactors were cooled to the reaction temperature and pressurized to the reaction pressure of 50 bar under Ar flow, the reaction mixture of CO (Messer, 3.0) $H_2$, and Ar with the desired composition was fed into each reactor at the desired volumetric flow rates, with the fraction of Ar fixed at 10 mol% as an internal standard.

Effluent streams were analyzed by gas chromatography (GC, Agilent 8890) every 2 h, with calculations performed taking average values of 3 measurements between ca. 8 and 12 h on stream. Response factors, $F_i$, for each compound $i$, respective to the internal standard Ar, in the GC analysis were determined by calibration using Eq. (1),

$$F_i = \frac{A_{Ar} / c_{Ar}^{in}}{A_i / c_i^{in}} \tag{1}$$

where $A_i$ is the integrated area determined for the peak of compound $i$ measured by the respective detector, and $c_i^{in}$ is the corresponding known molar concentration in the sampled calibration mixture. Representative GC results are shown in Supplementary Fig. 14.

The unknown effluent molar flow of compound $i$, $n_i^{out}$, was determined using Eq. (2),

$$n_i^{out} = \frac{A_i \times F_i}{A_{Ar}} \times n_{Ar}^{out} \tag{2}$$

where $n_{Ar}^{out}$ is the known flow of Ar at the reactor outlet.

The overall CO conversion ($X_{CO}$) was calculated using Eq. (3),

$$X_{CO} = \frac{n_{CO,in} - n_{CO,out}}{n_{CO,in}} \tag{3}$$

where $n_{CO,in}$ and $n_{CO,out}$ are the inlet and outlet CO molar flows, respectively. The selectivity for product $i$ ($S_i$) was calculated using Eq. (4),

$$S_i = \frac{n_{i,out} \times N_{C,i}}{\sum (n_{i,out} \times N_{C,i})} \times 100\% \tag{4}$$

where $n_{i,out}$ and $N_{C,i}$ are the molar flow and the number of carbon atoms in product $i$, respectively. The selectivity to higher alcohols and hydrocarbons were obtained summing the individual selectivity to alcohols and hydrocarbons with two or more carbon atoms. The space time yield of higher alcohols ($STY_{HA}$) expressed in $g_{HA}$ h$^{-1}$ $g_{cat}^{-1}$ was calculated using Eq. (5),

$$STY_{HA} = \sum \frac{n_{j,HA} \times MW_{j,HA}}{m_{cat}} \tag{5}$$

where $m_{cat}$ is the mass of the catalyst and $MW_{j,HA}$ is the molecular weight of higher alcohols containing $j$ carbon atoms.

The carbon balance ($\delta_C$) was determined according to Eq. 6 and showed always less than a 5% difference between the reactant and product streams.

$$\delta_C = \frac{n_{CO,in} - \sum n_{i,out} \times N_{C,i}}{n_{CO,in}} \cdot 100\% \tag{6}$$

### Catalyst characterization

X-ray fluorescence spectroscopy (XRF) was performed with an Orbis Micro-EDXRF spectrometer equipped with a Rh source and a silicon drift detector operated at 35 kV and 500 µA.

X-ray diffraction (XRD) was conducted using a Rigaku SmartLab diffractometer with a D/teX Ultra 250 detector using Cu K$\alpha$ radiation ($\lambda = 0.1541$ nm) and operating in a Bragg-Brentano geometry. Data were acquired in the 5−80°2$\theta$ range with an angular step size of 0.025° and a counting time of 1.5 s per step.

Temperature-programmed reduction with hydrogen ($H_2$-TPR) was performed at ambient pressure using a Micromeritics AutoChem HP II analyzer. Samples were loaded into a quartz tube, dried at 423 K in Ar for 1 h (10 K min$^{-1}$), and cooled to 313 K (20 K min$^{-1}$) in Ar. The temperature-programmed reduction was then carried out using 5 vol.% $H_2$ in $N_2$ (Messer) and increasing the temperature to 1073 K (5 K min$^{-1}$) with $H_2$ consumption quantified by a thermal conductivity detector.

Temperature-programmed $H_2$-$D_2$ exchange was performed using a Micromeritics AutoChem HP II analyzer coupled to a Pfeiffer OMNIStar mass spectrometer (MS). Used samples were loaded into a quartz tube, dried at 423 K in Ar for 1 h (10 K min$^{-1}$), and cooled to 313 K (20 K min$^{-1}$) in Ar. The $H_2$-$D_2$ exchange reaction was then carried out under 10 vol.% $D_2$ (Sigma-Aldrich, 99.8%) and 5 vol.% $H_2$ in $N_2$ (Messer) and increasing the temperature to 873 K (5 K min$^{-1}$), with the outlet flow of $H_2$, $D_2$ and HD quantified by online MS.

High-angle annular dark-field scanning transmission electron microscopy (HAADF-STEM) coupled to energy-dispersive X-ray (EDX) spectroscopy and elemental mapping were conducted using a Talos F200X instrument equipped with four silicon drift detectors (SDD) operated at an acceleration potential of 200 kV.

X-ray photoelectron spectroscopy (XPS) was performed using a Physical Electronics (PHI) Quantum 2000 X-ray photoelectron spectrometer featuring monochromatic Al K$\alpha$ radiation, generated from an electron beam operated at 15 kV and 32.3 W and a hemispherical capacitor electron-energy analyzer equipped with a channel plate and a position-sensitive detector. Analyses were conducted at $2 \times 10^{-7}$ Pa, with an electron takeoff angle of 45°, operating the analyzer in the constant pass energy mode. Prior to the measurements, used samples

diluted with quartz sand were ground and pressed firmly onto indium foil. The energy scale of the instrument was calibrated using Au and Cu as reference samples.

Diffuse reflectance infrared Fourier transform spectroscopy (DRIFTS) was performed using a Bruker Invenio S FT-IR spectrometer. The pre-reduced sample was mixed with $\alpha$-$Al_2O_3$ in a 1:2 mass ratio and inserted into the cell, subjected to in situ reduction in flowing $H_2$/He (50 vol.%) at 623 K for 2 h, and cooled down to 493 K in He. After pressurizing the cell to 20 bar under He flow, the background was collected once the pressure and IR signal stabilized. Continuous spectra at 1-min intervals with a resolution of 4 $cm^{-1}$ were collected upon switching to the $CO$/$H_2$ reaction mixture.

## Gaussian process and Bayesian optimization

As the mathematical relationship between the catalyst composition and reaction conditions to the catalyst performance metrics ($STY_{HA}$ and/or $S_{CO_2 + CH_4}$) is unknown, the optimization procedure is treated as a black-box function. In this study, Gaussian processes (GP) and Bayesian optimization (BO) were utilized to solve such black-box functions. We opt for a BO with GP surrogate model for the following reasons: (i) the input features and target variable are continuous in nature, (ii) implementation of GP-BO is less sensitive to the initial choice of hyperparameter, (iii) BO is known to require smaller initial datasets and fewer iterations to reach the optimal than genetic algorithms, and (iv) the uncertainty of GP surrogate model enables effective trade-off selection between exploration and exploitation, which can eventually foster efficient sampling of the chemical and parametric space of the catalyst[23,25,46].

A GP is a non-parametric and probabilistic model that defines a distribution over possible functions on the observed data, as described in Eq. (7),

$$f(x) \sim GP(m(x), k(x, x')) \tag{7}$$

where $m(x)$ is the mean function and $k(x, x')$ is the covariance function.

The mean and covariance kernel function in the Gaussian process was chosen and tuned during the model building process. Common kernel functions include constant, linear, square exponential, etc. We assume a constant mean function for simplicity and choose the squared exponential kernel to capture the smooth functions sampled by the GP, according to Eqs. (8)–(9),

$$m(x) = c \tag{8}$$

$$k_{SE}(x, x') = \sigma^2 \exp\left(\frac{-\frac{1}{2}(x - x')^2}{l^2}\right) \tag{9}$$

where $c$ is a constant, $\sigma^2$ is the variance parameter, $l$ is the lengthscale parameter and $(x - x')^2$ is the squared Euclidean distance between the two input datapoints. Details of the lengthscale parameters and their optimal values are presented in Supplementary Note 7 and Supplementary Table 30, respectively.

The acquisition function of BO is a key attribute used to guide the selection of the optimal set of solutions by balancing exploration (sampling regions where the objective function is uncertain) and exploitation (sampling regions where the objective function is likely to be optimal). Depending on single or multi-objective optimization, and to strike a balance between exploration and exploitation, various acquisition functions can be selected, such as expected improvement (EI), predictive variance (PV), upper confidence bound (UCB), and expected hypervolume improvement (EHVI), etc[23,35,46].

The BO campaigns with local penalization was implemented to recommend a batch of 30 data points to align with the experimental setup. For single objective optimization in Phase 1 and 2, we use a combination of EI and PV as acquisition functions given by Eq. (10)-,

$$EI(x) = \mathbb{R}[\max(f(x^*) - f(x), 0)] \tag{10}$$

$$PV(x) = P(f(x) \geq f(x^* + \varepsilon)) \tag{11}$$

where $x$ is the point at which the objective function is evaluated, $f(x)$ is the predicted mean of the objective function at point $x$, $f(x^*)$ is the best observed value, $\mathbb{R}$ denotes the expected value, and $\varepsilon$ is a small positive constant, that controls model improvement.

In Phase 3, the EHVI acquisition function was selected for multi-objective optimization, as described by Eq. (12),

$$EHVI(x) = \mathbb{R}[\max(H(D \cup \{x\}) - H(D), 0)] \tag{12}$$

where $x$ is the point at which the objective function is evaluated, $H(D)$ is the hypervolume of the dominated space by the current Pareto front $D$, $H(D \cup \{x\})$ is the hypervolume of the dominated space when $x$ is added to the current Pareto front, and $\mathbb{R}$ denotes the expected value. We highlight that the configuration of multi-objective optimization herein holds distinctive features from those reported in literature[35,37,54,59]. In our work, target variables are optimized by simultaneously varying the chemical composition and reaction conditions of the catalysts, which is computationally expensive and experimentally challenging. Another aspect is the indirect approach of the multi-objective optimization. Despite the initial goal being to simultaneously maximize $STY_{HA}$ and $S_{HA}$, the modest improvements to the latter, likely induced by intrinsic limitations of these catalysts, motivated us to minimize $S_{CO_2 + CH_4}$ (undesired products), which is an indirect approach to improve overall selectivity to valuable products, including higher alcohols, alkanes, and olefins. However, devising such strategies requires mechanistic understanding of higher alcohols synthesis and product distribution, which were effectively incorporated to identify several Pareto-optimal FeCoCuZr catalysts. Details of the model architecture and hyperparameter tuning for the GP-BO algorithms along with the optimal hyperparameters in each active learning cycle across all the three Phases are presented in Supplementary Note 7 and Supplementary Table 30.

## Implementation of the active learning loop

The generic workflow of the active learning across the various phases in this study followed a systematic sequence as described below:

(i) an initial set of experiments is planned and a series of catalysts with unique formulations is synthesized.

(ii) these catalysts are subjected to varying reaction conditions and their catalytic performance measured in terms of $STY_{HA}$, $S_{HA}$, $X_{CO}$, etc., are evaluated,

(iii) the experimental data (i.e., XRF-measured catalyst composition and reaction conditions) are used to train a GP regressor model, such that it predicts the performance metrics of interest (e.g., $STY_{HA}$)

(iv) the output of the GP prediction feeds to a BO algorithm, which in turn recommends the next set of experiments worth investigating. For each cycle Phases 1 and 2, a list of 30 catalytic systems is generated by using the EI function. This is followed by the generation of separate list of 30 candidates by using the PV function. In Phase 3, the EHVI function was used to generate 30 recommendations per cycle.

(v) The combined set of recommended catalysts from the step (iv) are manually screened and 6 catalysts are selected worth experimentation based on human-decision making the following criteria: high $STY_{HA}$ yielding catalysts (based on EI), diverse catalyst composition and reaction condition (based on PV), and avoid repetition from previous cycles as much as possible.

(vi)   the steps (i)-(v) are iteratively performed until the performance metric(s) of interest is achieved or no further improvements are observed, eventually leading to the stop criterion.

## Model quantification

To evaluate model performance, the predicted values of $STY_{HA}$ and $S_{CO_2 + CH_4}$ were compared with their respective experimentally measured counterparts, and the $R^2$, $RMSE$, and $MAPE$ values were calculated using Eqs. (13)–(15). Conceptually, high values of $R^2$ and low values of $RMSE$ and $MAPE$ indicate high model accuracy[57,60].

$$R^2 = 1 - \frac{\sum_{i=1}^{n}(y_{pred} - y_{act})^2}{\sum_{i=1}^{n}(y_{pred} - y_{mean})^2} \quad (13)$$

$$RMSE = \sqrt{\frac{1}{n}\sum_{i=1}^{n}(y_{pred} - y_{act})^2} \quad (14)$$

$$MAPE = \frac{1}{n}\sum_{i=1}^{n}\frac{y_{pred} - y_{act}}{y_{act}} \quad (15)$$

where $y_{pred}$ and $y_{act}$ are the predicted and measured values of the target variable, respectively, $y_{mean}$ is the mean of all measured values of the target variable, and $n$ is the total number of datapoints.

In addition to experimental verification, the evaluation of the predictive reliability of the Gaussian process model was performed by cross-validation at the end of each Phase. Given the small sample size i.e., 30, 36 and 18 datapoints, generated from the active learning loop, in Phases 1, 2, and 3 respectively, the stratified cross-validation technique was applied for this task. (See Supplementary Note 8 and Supplementary Table 31 for an extended discussion).

## Composition-productivity relationship

The unsupervised $k$-means algorithm was used to investigate the effect of varying composition on $STY_{HA}$, such that all the catalysts synthesized and tested in Phase 1 were grouped into distinct clusters. This algorithm identifies homogeneous subgroups of features in a given dataset[60]. In an $n$-dimensional space with $m$ features, the algorithm divides the features into $k$ clusters (where $k \leq n$), minimizing the sum of squares within each cluster. Similarity between features within a cluster is determined by their distance, commonly measured using Euclidean distance. The objective function of $k$-means is expressed in Eq. (14) and the optimal number of $k$ clusters was identified using the Elbow method.

$$J = \sum_{j=1}^{k}\sum_{i \in c_j}(x_i - c_j)^2 \quad (16)$$

where $J$ is the objective function, $x_i$ is the $i^{th}$ datapoint, $c_j$ is $j^{th}$ cluster centre (centroid), and $k$ is the number of clusters.

## Feature-importance analysis

We employed a posteriori feature-importance using the SHAP (SHapley Additive exPlanations) on the GP algorithm to quantitatively estimate the contributions of each input feature on the model predictions, referred as interpretability. For each experimental datapoint, this methodology perturbs the features and determines importance value for every prediction. Furthermore, by averaging SHAP values across the entire dataset, the overall importance of input features is determined. As such, it facilitates local and global interpretability, respectively, with higher-ranked features exhibiting larger SHAP values and vice versa[56–58].

All modeling activities were performed in Python (version 3.6), using libraries including Pandas 2.0.3 and NumPy 1.24.3 for data analysis, scikit-learn 1.3.0 for performing $k$-means clustering and data analysis, GPflow 2.9.0 and Trieste 1.2.0 for implementing the GP and BO algorithms, respectively and Shap 0.40.0 for model interpretation.

## Data availability

The curated dataset generated in this study have been deposited in the Zenodo database (https://doi.org/10.5281/zenodo.11113829). Further data supporting the findings of this study are provided in the Supplementary Information and Source Data file.

## Code availability

The curated dataset and the Python codes for data pre-processing and fine-tuning the Bayesian optimization along with necessary instructions to run the model can be found on GitHub at https://github.com/ssuvarnamanu/active-learning-for-HAS.

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

## Acknowledgements

This study was created as part of NCCR Catalysis (grant number 180544), a National Centre of Competence in Research funded by the Swiss National Science Foundation and was supported by Sulzer Chemtech AG. T.Z. thanks the Agency for Science, Technology and Research (A*STAR) Singapore for support through a graduate fellowship. The Scientific Center for Optical and Electron Microscopy (ScopeM) at ETH Zurich is thanked for access to their facilities. Dr. D. Faust Akl, Dr. F. Krumeich, and V. Giulimondi are acknowledged for STEM-EDX and XPS measurements. We are grateful to Dr. P. Preikschas and A. Nabera for fruitful discussions on the manuscript and assistance with environmental and economic cost calculations. C. Ko is thanked for help with illustrations.

## Author contributions

M.S., T.Z., A.J.M., and J.P.R. conceptualized the stages of this project. M.S. developed the software codes and performed data analysis with assistance from S.H.C. T.Z. and S.H.C. synthesized the catalysts and performed the catalytic tests. T.Z. and Y.G. conducted catalyst

characterization experiments and analyzed the data. A.J.M supervised the work and reviewed the manuscript. J.P.R. conceived and supervised the entire project, reviewed the manuscript, and managed resources and funding. All the authors provided inputs to the manuscript and approved the final version.

## Funding

## Competing interests
The authors declare no competing interests.
