## [Peer Review File · Nature Communications]

Active learning streamlines development of high performance catalysts for higher alcohol synthesisREVIEWER COMMENTS

Reviewer #1 (Remarks to the Author):

Developing efficient catalysts for syngas-based higher alcohol synthesis (HAS) remains a formidable research challenge. This work presents an alternative strategy by integrating active learning into experimental workflows, exemplified via the FeCoCuZr catalyst family. The data-aided framework streamlined the navigation of extensive composition and reaction condition space in 86 experiments, offering >90% reduction in environmental footprint and costs over traditional programs. It identified the Fe₆₅Co₁₉Cu₅Zr₁₁ catalyst with optimized reaction conditions to attain higher alcohol productivities of 1.1 g_{HA} h⁻¹ g_{cat}⁻¹ under stable operation for 150 h on stream, a 5-fold improvement over typically reported yields. However, few of the points may be clarified for the benefit of the readers. My views are appended below:

1. In the abstract the authors have claimed an active learning of the FeCoCuZr catalyst family, I suggest a detailed statement of the reasons for the choice of this catalyst.
2. In Fig. 3a the structural characterization of the Fe₆₉Co₁₂Cu₁₀Zr₉ and Fe₇₉Co₁₀Zr₁₁ catalyst was exhibited, the ICP-AES results of the elemental contents should be given to confirm the successful synthesis of the target catalyst.
3. The authors reported Fe₆₉Co₁₂Cu₁₀Zr₉ and Fe₇₉Co₁₀Zr₁₁. A simple comparison of the TOF values for these catalysts should be added for the reader's convenience.
4. The catalyst design guidelines in Fig. 6 are based on iterative learning over FeCoCuZr catalysts, so whether they can be analyzed prospectively as a HAS system is to be considered. This should be clarified through discussion.
5. It is recommended to discuss the existence of metallic carbide species. Is there any relationship between the relative ability of CO non-dissociated activation and metallic carbide species?
6. It is suggested to perform the FTIR spectrum to investigate the reaction intermediates on the surface of FeCoCuZr catalysts.
7. With the reaction time prolonging, the structural change of used catalyst during the activation reaction process should be compared to fresh catalyst, the authors should give an experiment for this phenomenon.

Reviewer #2 (Remarks to the Author):

This is an interesting and timely paper describing multi-objective optimization to develop catalysts for higher alcohol synthesis. I think the manuscript provides new insights to the field and would be appreciated by the community. The paper can be accepted after following suggestions/comments are addressed.

- 1) Please describe the ML analysis methods more clearly. For instance, I could not find the information on how the authors tuned hyperparameters.
- 2) Related to the question above, I could not find information about predictive reliability of the ML model. I understand that the authors check this by performing experimental validation. But, cross validation within their datasets at each phase should be provided to assess the reliability of the ML model.
- 3) Multi-objective optimization itself is a widely known technique, used in fields including MI and also catalysis, to find Pareto fronts (e.g. Nature Communications, 2022, 13, 995; Adv. Mater. 2023, 35, 2211497; Nature Chemical Engineering, 2024, 1, 240–250). In comparison to these methods, are there any improvements/merits of the approach proposed in this study? Please describe methodological aspects more clearly.
- 4) The authors are encouraged to compare catalytic performance with literature value and/or state-of-the-art catalytic systems to clearly show the developed system is unique and outstanding.

5) Product selectivity needs to be properly reported. Specifically, details regarding the types of higher alcohols (HAs), paraffins, and olefins produced should be clearly described.

Reviewer #3 (Remarks to the Author):

(These remarks are confined to the active learning aspects of the paper, we defer to other reviewers for the specific technical aspects of the catalyst characterization.)

The authors present an active learning workflow to realize high productivity while also reducing carbon footprint and costs comparing to traditional campaigns. They thoroughly review the recent literature on ML-guided experiments for catalysis. The active learning approach follows current best practices, is well-explained for non-experts, and is effective for their problem. The economic and carbon emission evaluation of the process is quite interesting and presents a compelling justification for using ML-guided experimentation. In summary, the work is sound and impactful. We have no major criticisms of the paper. Below are some *minor* comments/suggestions for ways in which the presentation could be clarified to benefit readers:

- Line 122 and Line 499 both talk about PV but give different explanations as predictive variance and Probability of Improvement. Making this definition consistent may be clearer.
- Line 142 talking about clustering: Would it be better to explain why SHA was chosen for the x-axis?
- Line 126 implies EI for exploitation; however, EI and PV are both acquisition functions that are designed to balance exploitation and exploration. Is the EI used here a modified function that is more concerned about exploitation and PV more concerned about exploration?
- Line 128 talking about the role of human decision-making in providing a judgement and selection from the suggested compositions. The text suggests that a human selected a combination of candidates from the EI and PV candidate lists. That is fine, but could this be explained a bit more? What were the humans selecting for? (Or stated more adversarially, would the humans have reached this answer anyway, if provided with some random compositions to select from?)
- Line 49, is it better to include how $m(x)$ and $k(x, x')$ calculated?
- General comment on math typesetting: Some of the equations in the manuscript (.docx) and SI (PDF) appeared with unusual accented symbols when viewed on a Mac; but this may be an artifact of our computers (we tried on two different computers). For example, Line 505-512, equations have some visualization problem.
- Line 503 indicates use a combination of EI and PV, how this combination work? (Is it a sum or multiplication of these acquisition functions? How is the small value ϵ being chosen?)
- Re: Figure 5a- left: The Pareto front is depicted as a dashed diagonal line between the red and blue points; but given the non-dominating nature of the Pareto front, shouldn't it also include the unfilled circles to the left of the line?
- Re: Figure 5a-right: One of the key results claimed by the authors is that they extended the Pareto front of performance through their active learning cycle (purple and green points to the left of the previous dashed Pareto front line). But these improvements appear quite modest in this plot.

- The authors include the data for the different iterations of their experiment Tables S1-S20 (great!). But this is unrelated to the data on their github repository (https://github.com/ssuvarnamanu/active-learning-for-HAS/blob/main/HAS_model_data_all.xlsx). The latter is much smaller and has different data points that do not correspond to the results in Tables S1-S20. Presumably it is sample data or some kind for program testing? It would be helpful if the complete dataset (along with the phase & cycle information) were provided in a machine-readable format in their github repository, to facilitate reanalysis of the data.
- In a similar vein, the code on their github repository appears to be an example, rather than code that can be used to replicate the results shown in the paper. Again, including this would facilitate reproducing their work.
- The code uses relatively standard python packages. However, for reproducibility, the authors should provide version numbers for the packages that they use (seabird, tensor flow, sci-kit, gpflow, etc.). Depending on the particular python package manager they are using, they should provide a configuration file which specifies the necessary dependencies and version information (e.g., setup.yaml or environment.yml file)
- It would be helpful to clarify in the data availability statement on p.24 that the provided link <https://doi.org/10.5281/zenodo.1030139> only contains the ecological impact data.

Reviewer #3 (Remarks on code availability):

As noted in our comments for the authors, the code and data provided appear to be a sample or demonstration, rather than the actual data/code used for the study.

Additionally, they can improve the instructions for installation (and improve reproducibility) by providing version information, as noted in our comments.

Reviewer #4 (Remarks to the Author):

Response to Reviewers for Manuscript NCOMMS-24-14026

Code: Comments in blue | Replies in black | Actions in bold

Note: Indicated page, line, figure, or reference numbers refer to the revised manuscript and/or supplemental information with changes highlighted.

Reviewer #1

Developing efficient catalysts for syngas-based higher alcohol synthesis (HAS) remains a formidable research challenge. This work presents an alternative strategy by integrating active learning into experimental workflows, exemplified via the FeCoCuZr catalyst family. The data-aided framework streamlined the navigation of extensive composition and reaction condition space in 86 experiments, offering >90% reduction in environmental footprint and costs over traditional programs. It identified the $\text{Fe}_{65}\text{Co}_{19}\text{Cu}_5\text{Zr}_{11}$ catalyst with optimized reaction conditions to attain higher alcohol productivities of $1.1 \text{ g}_{\text{HA}} \text{ h}^{-1} \text{ g}_{\text{cat}}^{-1}$ under stable operation for 150 h on stream, a 5-fold improvement over typically reported yields. However, few of the points may be clarified for the benefit of the readers. My views are appended below:

We thank the Reviewer for appreciating the breadth and significance of our study and their valuable suggestions and critical feedback. These suggestions helped us to further improve the impact of our piece, as detailed below.

1. In the abstract the authors have claimed an active learning of the FeCoCuZr catalyst family, I suggest a detailed statement of the reasons for the choice of this catalyst.

As briefly mentioned in the Introduction, the choice of studying the FeCoCuZr catalyst family stems from the previous discovery of highly active ZrO_2 -promoted bimetallic catalysts for HAS from syngas (*ACS Catal.* **2023**, 13, 9946-9959). It was found that small contents of ZrO_2 (ca. 5-20 mol%) generally resulted in a promotional effect (increased higher alcohol productivities and selectivities) for Fe-Co, Fe-Cu, and Co-Cu catalysts regardless of the bimetallic pairing and the ratio of the two active metals. Furthermore, bimetallic catalysts always showed superior catalytic performance compared to monometallic counterparts containing only one of the two active metals, showing the positive effect of the synergy between active metals on higher alcohol formation. As such, it would be reasonable to assume that combining the four elements Fe, Co, Cu, and Zr in appropriate ratios would result in even more productive catalysts by maximizing their interactions. The large resulting parameter spaces of compositions and reaction conditions and the lack of previous experience in literature with these materials (and thus lack of guidelines) made this family an outstanding candidate for the application of data-driven catalyst development. **We have strengthened the Introduction of the revised manuscript (page 4, lines 74-80) with the above justification.**

2. In Fig. 3a the structural characterization of the $\text{Fe}_{69}\text{Co}_{12}\text{Cu}_{10}\text{Zr}_9$ and $\text{Fe}_{79}\text{Co}_{10}\text{Zr}_{11}$ catalyst was exhibited, the ICP-AES results of the elemental contents should be given to confirm the successful synthesis of the target catalyst.

We clarify that the catalyst compositions indicated as subscripts ($\text{Fe}_x\text{Co}_y\text{Cu}_z\text{Zr}_a$) throughout the manuscript, as well as those presented in Supplementary Tables 1-7, 9, 11-16, 18-20, are actual compositions as measured by XRF and were used as model inputs. To avoid confusion, **we have clarified**

in the Methods section (page 24, line 595) that only actual XRF-measured compositions are presented and used as inputs in active learning. Furthermore, we have included suggested compositions for each active learning cycle in addition to measured ones in full Excel dataset uploaded to the Zenodo repository, showing always a close match within 5% between suggested and actual compositions for each catalyst.

Nonetheless, we have performed additional ICP-AES measurements on the target $\text{Fe}_{69}\text{Co}_{12}\text{Cu}_{10}\text{Zr}_9$ catalyst to further ensure correctness of its composition and the reliability of the XRF measurements. The results are shown below in **Table R1**, confirming the successful synthesis of the targeted composition and the accuracy of XRF-measured values in this work.

Table R1. Suggested and actual compositions of the target $\text{Fe}_{69}\text{Co}_{12}\text{Cu}_{10}\text{Zr}_9$ catalyst as measured by XRF and ICP-AES.

Composition	Fe (mol%)	Co (mol%)	Cu (mol%)	Zr (mol%)
Suggested	69	11	11	9
Measured (XRF)	68.7	11.7	10.2	9.4
Measured (ICP-AES)	69.6	11.0	11.2	8.2

3. The authors reported $\text{Fe}_{69}\text{Co}_{12}\text{Cu}_{10}\text{Zr}_9$ and $\text{Fe}_{79}\text{Co}_{10}\text{Zr}_{11}$. A simple comparison of the TOF values for these catalysts should be added for the reader's convenience.

In contrast to simpler reactions catalyzed by supported catalysts, where all surface atoms of metal nanoparticles can be considered to be active in forming a single product, calculation of TOFs is highly challenging for higher alcohol synthesis catalysts, and specifically over FeCoCuZr catalysts, due to the high complexity of the reaction and catalyst surface. Four metals in the form of metallic, oxidic, and carbide species are present, simultaneously catalyzing CO dissociation, non-dissociative CO insertion, hydrogenation, and termination steps. Furthermore, it is not possible presently to measure or estimate the number of active sites, as the nature of active ensemble(s) is not yet clear.

Nevertheless, we acknowledge the value of a standardized comparison among materials. We estimated the rate of formation of each alcohol per unit area of the used catalyst surface (**Table R2**), which has a similar function to TOF values. By this measure, the $\text{Fe}_{69}\text{Co}_{12}\text{Cu}_{10}\text{Zr}_9$ discovered displays a superior activity in the formation of each alcohol compared to the $\text{Fe}_{79}\text{Co}_{10}\text{Zr}_{11}$ benchmark by a factor of 2. We have added the data in **Table R2** to the revised Supplementary Information as **Supplementary Table 10**, and mentioned the comparison in the revised manuscript (page 8, lines 194-195).

Table R2. Rate of alcohol formation per unit area of $\text{Fe}_{69}\text{Co}_{12}\text{Cu}_{10}\text{Zr}_9$ and $\text{Fe}_{79}\text{Co}_{10}\text{Zr}_{11}$ catalysts.

Catalyst	$S_{\text{BET}} / \text{m}^2 \text{g}_{\text{cat}}^{-1}$		Rate of alcohol formation / $\text{mmol h}^{-1} \text{m}_{\text{cat}}^{-2}$			
	Fresh	Used	C ₁	C ₂	C ₃	C ₄₊
$\text{Fe}_{79}\text{Co}_{10}\text{Zr}_{11}$	146	38	0.20	0.12	0.03	0.01
$\text{Fe}_{69}\text{Co}_{12}\text{Cu}_{10}\text{Zr}_9$	136	24	0.39	0.24	0.07	0.02

4. The catalyst design guidelines in Fig. 6 are based on iterative learning over FeCoCuZr catalysts, so whether they can be analyzed prospectively as a HAS system is to be considered. This should be clarified through discussion.

We thank the Reviewer for raising this important point. The guidelines presented in Figure 6 were indeed formulated after all three phases of the entire active learning process, following the collection of all data for each cycle. As such, the quantitative findings suggested by the guidelines (e.g. operating temperature range for Pareto-optimal catalysts) and compositional guidelines would apply only to the FeCoCuZr catalyst family studied and would arguably not be directly relevant in designing HAS systems with different active metals, promoters, and architectures. However, we note from our experience and literature that qualitative findings, especially those relating to operating conditions (e.g. Pareto-optimal catalysts requiring intermediate temperatures and high GHSV) are likely not dependent on catalyst architecture and synthesis method, and may apply to prospective HAS from syngas catalysts in general. Furthermore, we are confident that this active learning methodology and retroactive formulation of performance guidelines from data can be applied to other reactions and thus recommend other users formulate guidelines specific to their application during the active learning process. **We have added this discussion to the revised manuscript (page 15, lines 367-376).**

5. It is recommended to discuss the existence of metallic carbide species. Is there any relationship between the relative ability of CO non-dissociated activation and metallic carbide species?

While metallic carbide species have long been recognized as active species in HAS, proving a direct link between the presence of such species to the ability of the catalyst surface to activate CO non-dissociatively is not trivial. Many works find a correlation between elevated proportion of carbides and improved S_{HA} , including in our previous work on ZrO₂-promoted bimetallic catalysts (*ACS Catal.* **2023**, *13*, 9946-9959), where the increased rate of alcohol formation requires non-dissociative CO activation to be enhanced. Furthermore, mechanistic roles of the interface between Fe carbide species and partially reduced Fe oxides in non-dissociative CO activation has been postulated in the literature, largely based on computational calculations (*Appl. Catal. B* **2022**, *307*, 121155, *Appl. Catal. B* **2022**, *312*, 121393).

Recently in a separate study from our group, we have been able to use CO-TPD to distinguish surface sites of Fe₉₀Zr₁₀ catalysts based on their relative CO adsorption strength, discovering that the proportion of binding sites corresponding to the interface between Fe₅C₂, Fe₃O₄, and ZrO₂ exhibit a strong correlation with STY_{HA} and S_{HA} , (*Chem. Catal.* **2024**, manuscript accepted). While such experimental analyses are in principle possible for FeCoCuZr catalysts, we anticipate challenges in the differentiation of surface sites due to a larger variety of species by tripling the number of active metals. We envision that such a dedicated characterization study would be useful in establishing structure-mechanism relationships but currently falls outside the scope of this work. **We have additionally cited the above works and strengthened the explanation of the role of the interface between metallic carbide/oxide species in the revised manuscript (page 8, lines 174-176).**

6. It is suggested to perform the FTIR spectrum to investigate the reaction intermediates on the surface of FeCoCuZr catalysts.

We thank the Reviewer for this valuable suggestion. We performed FTIR spectroscopy, specifically *in situ* diffuse reflectance infrared Fourier transform spectroscopy (DRIFTS) of the optimal Fe₆₉Co₁₂Cu₁₀Zr₉ catalyst under relevant reaction conditions, with the results shown in **Figure R1**. Signals corresponding to *CH_x (*CH₂ or *CH₃), CH₃O* and CH₃CH₂O* intermediates were detected, in line with the expected

mechanisms of *CH_x coupling and non-dissociative *CO insertion required to form the ethoxy intermediate. Additionally, gas-phase CH_4 and CO_2 (not shown due to large intensity) were detected as products formed. Other oxygenates detected (*CO_3 , $HCOO^*$, and CH_3CHO^*) could have been a result of adsorption and hydrogenation of CO_2 . These findings are discussed in the revised manuscript (page 9, lines 198-200) and we have added Figure R1 as Supplementary Figure 6.

Figure R1. *In situ* DRIFT spectra for the $Fe_{69}Co_{12}Cu_{10}Zr_9$ catalyst. Reaction conditions: $H_2:CO = 2.0$, $T = 533\text{ K}$, $P = 20\text{ bar}$, $m_{cat} = 10\text{ mg}$, $F_T = 7.5\text{ cm}^3\text{ min}^{-1}$, and dwell time = 180 min.

7. With the reaction time prolonging, the structural change of used catalyst during the activation reaction process should be compared to fresh catalyst, the authors should give an experiment for this phenomenon.

To study the structural change of the $Fe_{69}Co_{12}Cu_{10}Zr_9$ catalyst during the pre-treatment and reaction processes, we have additionally performed XRD and N_2 sorption measurements on the fresh, reduced (following pre-treatment under dilute H_2), and used (following exposure to reaction conditions for 24 h and 100 h) catalysts. This shows the evolution of average Fe_3O_4 crystallite size ($d_{Fe_3O_4}$, calculated using the Scherrer equation using the $Fe_3O_4(511)$ peak) and BET surface area (S_{BET}), as presented in **Table R3**.

Table R3. Structural properties of $Fe_{69}Co_{12}Cu_{10}Zr_9$ at various stages of the reaction.

Status	TOS / h	$d_{Fe_3O_4} / \text{nm}$	$S_{BET} / \text{m}^2\text{ g}_{cat}^{-1}$
Fresh	-	9	136
Reduced	0	9	52
Used	24	15	24
Used	100	15	20

We have performed additional STEM-EDX measurements of $Fe_{69}Co_{12}Cu_{10}Zr_9$ as calcined and after 24 h of reaction showing the drastic structural change. These images are displayed in the **Supplementary Figure 3**. XRD patterns and STEM-EDX images of these catalysts are also compared in **Figure R2**. Overall, the greatest structural change occurs during reductive pre-treatment and during the initial hours of reaction, as S_{BET} sharply decreases with the reduction of bulk Fe_3O_4 and CuO to partially reduced Fe phases and Cu. Dispersed Cu in the fresh catalyst as calcined notably agglomerates after

reduction and the initial stage of reaction. Remarkably, no significant change in either $d_{\text{Fe}_3\text{O}_4}$ or S_{BET} was apparent between 24 and 100 h, highlighting the remarkable stability of the catalyst after equilibration. This discussion has been incorporated into the revised manuscript (pages 8, lines 177-184). We have also added the data in Table R2 to the revised Supplementary Information as Supplementary Table 9, as well as additional XRD patterns to Supplementary Figure 5, respectively.

Figure R2. STEM-EDX elemental maps of the $\text{Fe}_{69}\text{Co}_{12}\text{Cu}_{10}\text{Zr}_9$ catalyst (a) in fresh form, and after (b) 24 h and (c) 100 h of use in CO hydrogenation. Scale bars: 50 nm. (d) XRD patterns of the $\text{Fe}_{69}\text{Co}_{12}\text{Cu}_{10}\text{Zr}_9$ catalyst at various stages of the reaction. Reaction conditions: $\text{H}_2:\text{CO} = 2.0$, $T = 533 \text{ K}$, $P = 50 \text{ bar}$, and $GHSV = 24,000 \text{ cm}^3 \text{ h}^{-1} \text{ g}_{\text{cat}}^{-1}$.

Reviewer #2

This is an interesting and timely paper describing multi-objective optimization to develop catalysts for higher alcohol synthesis. I think the manuscript provides new insights to the field and would be appreciated by the community. The paper can be accepted after following suggestions/comments are addressed.

We appreciate the Reviewer's positive feedback, recognizing the timeliness and relevance of our study to the broad audience of both experimental and theoretical catalysis practitioners. Their valuable recommendations have helped to improve the quality of our study, as detailed below.

1. Please describe the ML analysis methods more clearly. For instance, I could not find the information on how the authors tuned hyperparameters.

We thank the Reviewer for this pertinent comment on the core of this study. For tuning the hyperparameters of the Gaussian process regressor, we selected the Squared Exponential kernel, and initialized the input specific lengthscales (hyperparameters of the algorithm) to a default value of 0.1, and optimized them using the Scipy optimizer to minimize the negative log likelihood under constant variance 1.0. This methodology enabled automated selection of kernel lengthscales ensuring effective model performance. **We have now presented a more detailed description of the model architecture and its hyperparameters in the Supplementary Note 7, and Supplementary Table 30.**

2. Related to the question above, I could not find information about predictive reliability of the ML model. I understand that the authors check this by performing experimental validation. But, cross validation within their datasets at each phase should be provided to assess the reliability of the ML model.

As correctly pointed, we evaluated the robustness of the ML predictions through experimental validation, achieving significant overall accuracy as shown in Fig.4d of the manuscript. We believe this is the most robust approach to validate ML models and specifically in active learning campaigns for chemistry and catalysis-related applications. In attendance to the Reviewer's recommendation, we present cross-validation for the different Phases to demonstrate predictive reliability of the Gaussian process model. Since Phases 1, 2, and 3 have 30, 36 and 18 datapoints respectively, generated from the active learning loop, which is a small sample size, we employ the stratified cross-validation technique for this task. **Details of these results are presented in the, Supplementary Note 8, and Supplementary Table 31.**

3. Multi-objective optimization itself is a widely known technique, used in fields including ML and also catalysis, to find Pareto fronts (e.g. Nature Communications, 2022, 13, 995; Adv. Mater. 2023, 35, 2211497; Nature Chemical Engineering, 2024, 1, 240–250). In comparison to these methods, are there any improvements/merits of the approach proposed in this study? Please describe methodological aspects more clearly.

We must clarify that we do not claim any improvements or developments from an algorithmic perspective in this study since we have used the Gaussian-process as surrogate model with Bayesian optimizer catered to our application. However, in comparison to the above-mentioned studies and related ones in recent literature, our approach of utilizing multi-objective learning holds two distinctive features.

Firstly, we optimize target variables by simultaneously varying chemical composition and reaction conditions of the catalysts; adding a new layer of complexity. Previous studies in materials informatics and

catalysis have primarily focused only on either optimizing catalyst composition (*Adv. Mater.* **2023**, 35, 2211497), or reaction conditions (*Nat. Commun.*, **2022**, 13, 995; *Nat. Chem. Eng.* **2024**, 1, 240–250), as optimizing both these parameter classes can be computationally expensive and experimentally challenging. Furthermore, we use an indirect approach in our multi-objective optimization. Though our initial goal was to simultaneously maximize STY_{HA} and S_{HA} , the inability of the optimizer to make significant changes to the latter target due to the inherent mechanistic properties of the catalyst, motivated us to minimize $S_{CO_2+CH_4}$ (undesired products) which is an indirect approach to improve the productivity of higher alcohols. Importantly, devising such strategies requires domain knowledge by humans, which we effectively incorporated in the multi-objective optimization scheme to help us identify several Pareto-optimal HAS catalysts. Following the Reviewer's recommendation, **we discuss the merits of our multi-objective learning strategy in the revised manuscript (page 23, lines 574-583).**

4. The authors are encouraged to compare catalytic performance with literature value and/or state-of-the-art catalytic systems to clearly show the developed system is unique and outstanding.

Fig. 4c compares our systems with over 120 literature reported HAS catalysts. In the revised version, **we explicitly highlight this point and present this comparative discussion in the revised manuscript (page 11, lines 253-260).** The literature reported values of HAS systems along with the reference DOI of the publications are included in the Supplementary Source Data.

5. Product selectivity needs to be properly reported. Specifically, details regarding the types of higher alcohols (HAs), paraffins, and olefins produced should be clearly described.

We thank the Reviewer for raising this important point. Indeed, the original manuscript reported the combined selectivity towards all higher alcohols (C_2-C_6), paraffins (C_2-C_{10}), and olefins (C_2-C_8). **Due to space constraints, we now include full product distributions for each experiment listing the selectivities towards individual higher alcohols, paraffins, and olefins for every experiment across active learning cycles in the supplementary Excel sheet uploaded to Zenodo and GitHub. For the convenience of the reader, we include the full product distributions of the best performing catalyst of each cycle of Phase 1 and 2 in Supplementary Tables 8 and 20, respectively, and all catalysts of cycle 3 of Phase 3 in Supplementary Table 25.**

Reviewer #3

The authors present an active learning workflow to realize high productivity while also reducing carbon footprint and costs comparing to traditional campaigns. They thoroughly review the recent literature on ML-guided experiments for catalysis. The active learning approach follows current best practices, is well-explained for non-experts, and is effective for their problem. The economic and carbon emission evaluation of the process is quite interesting and presents a compelling justification for using ML-guided experimentation. In summary, the work is sound and impactful. We have no major criticisms of the paper. Below are some *minor* comments/suggestions for ways in which the presentation could be clarified to benefit readers:

We thank the Reviewer for their encouraging and positive review. As active learning is a nascent area in catalysis, we put emphasis on accessibility to various communities and underscored immediate advantages in terms of sustainability. We have carefully revised the manuscript following their recommendations to improve the presentation of our study.

1. Line 122 and Line 499 both talk about PV but give different explanations as predictive variance and Probability of Improvement. Making this definition consistent may be clearer.

We thank the Reviewer for highlighting the discrepancy. In the revised version, **we present a uniform definition for the acquisition function predictive variance (PV) and keep it consistent throughout the revised manuscript (page 6, line 129 and page 22, line 557).**

2. Line 142 talking about clustering: Would it be better to explain why S_{HA} was chosen for the x-axis?

The primary objective of Phase 1 was to maximize productivity by varying the catalysts composition. A cluster plot of the catalysts synthesized and tested in this phase, with STY_{HA} on the y-axis and S_{HA} on the x-axis allowed us to identify catalysts with high STY_{HA} and S_{HA} – both of which are key performance metrics. Based on this analysis we were also able to understand compositional trends driving performance as shown in **Figure 2c-d**, which helped us with informed decision making regarding experiment selection in subsequent phases. **We have explained the selection of S_{HA} on the x-axis in the revised manuscript (page 7, lines 153-157).**

3. Line 126 implies EI for exploitation; however, EI and PV are both acquisition functions that are designed to balance exploitation and exploration. Is the EI used here a modified function that is more concerned about exploitation and PV more concerned about exploration?

We appreciate the question raised by the Reviewer and realize that our initial description of the use of EI and PV acquisition functions was not clear enough. To clarify the same, we do not use any modified EI or PV function that distinctly enhances exploitation or exploration. We use these functions without any modification, generate a distinctive list of catalyst candidates in a batch-wise manner across cycles, and select the candidates worth experimentation based on human expertise. For example, in each cycle of Phase 1, we first generate a list of 30 candidates by using the EI function with the aim to maximize STY_{HA} . We then separately generate a list of 30 candidates by using the PV function to diversify the catalyst composition. The combined set of 60 recommended catalysts from the above steps are manually screened and 6 catalysts are selected worth experimentation based on the following criteria:

- high STY_{HA} yielding catalysts (based on EI),
- diverse catalyst composition (based on PV) and
- avoid repetition from previous cycles as much as possible.

We elaborate on this methodology in the revised manuscript (page 24, lines 599-614).

4. Line 128 talking about the role of human decision-making in providing a judgement and selection from the suggested compositions. The text suggests that a human selected a combination of candidates from the EI and PV candidate lists. That is fine, but could this be explained a bit more? What were the humans selecting for? (Or stated more adversarially, would the humans have reached this answer anyway, if provided with some random compositions to select from?)

This question is addressed together with Comment #3 by Reviewer#3. **Kindly refer to our previous response.**

5. Line 493, is it better to include how $m(x)$ and $k(x,x')$ calculated?

We have included the calculations of $m(x)$ and $k(x,x')$ for the Gaussian process regressor in the revised manuscript (pages 21-22, lines 543-549).

6. General comment on math typesetting: Some of the equations in the manuscript (.docx) and SI (PDF) appeared with unusual accented symbols when viewed on a Mac; but this may be an artifact of our computers (we tried on two different computers). For example, Line 505-512, equations have some visualization problem.

All the equations in the manuscript and the SI were prepared using the journal recommended software MathType and then copied to MS Word on a Windows operating system. We are aware that MathType generated equations encounter visualization problems occasionally when viewed on a Mac. However, based on our previous experience, **we ensure the MathTyped equations will be provided in journal compliant format while undertaking the proofs.**

7. Line 503 indicates use a combination of EI and PV, how this combination work? (Is it a sum or multiplication of these acquisition functions? How is the small value ϵ being chosen?)

This question is addressed together with Comment #3 by Reviewer#3. **Kindly refer to our previous responses.**

8. Re: Figure 5a- left: The Pareto front is depicted as a dashed diagonal line between the red and blue points; but given the non-dominating nature of the Pareto front, shouldn't it also include the unfilled circles to the left of the line?

We appreciate the Reviewer's inquiry. The Pareto front depicted as a dashed line was drawn connecting the highest STY_{HA} (blue) with the least $S_{CO_2+CH_4}$ (red) value. The points to the left of the line found during Phase 2 also qualify as Pareto optimal. To avoid confusion and to ensure conformity with non-dominating

nature of the Pareto front, we have included all the datapoints prior to the right of the dashed line, except the Pareto-optimal catalysts found in Phase 3 as shown in revised Figure 5a.

9. Re: Figure 5a-right: One of the key results claimed by the authors is that they extended the Pareto front of performance through their active learning cycle (purple and green points to the left of the previous dashed Pareto front line). But these improvements appear quite modest in this plot.

We agree with the Reviewer's comment. Our goal in multi-objective active learning was to find catalysts that break the Pareto barrier. Notably, although we find Pareto-optimal candidates, they are not significantly away from Pareto barrier and mostly situated on top of it. We duly acknowledge this fact, the reason for which is rooted in the intrinsic limitation of HAS catalysts to achieve $S_{\text{CO}_2+\text{CH}_4} < 30\%$ either through composition or reaction condition optimization. Furthermore, in the absence of quantifiable techno-economic data and community consensus on practically relevant productivities or selectivities for HAS, we believe $STY_{\text{HA}} = 0.65 \pm 0.05 \text{ g}_{\text{HA}} \text{ h}^{-1} \text{ g}_{\text{cat}}^{-1}$ and $S_{\text{CO}_2+\text{CH}_4} = 34 \pm 2\%$, as identified in this study could be a fair-trade off between these performance metrics. **We discuss these aspects in the revised manuscript (page 12-13, lines 303-316).**

10. The authors include the data for the different iterations of their experiment Tables S1-S20 (great!). But this is unrelated to the data on their github repository (https://github.com/ssuvarnamanu/active-learning-for-HAS/blob/main/HAS_model_data_all.xlsx). The latter is much smaller and has different data points that do not correspond to the results in Tables S1-S20. Presumably it is sample data or some kind for program testing? It would be helpful if the complete dataset (along with the phase & cycle information) were provided in a machine-readable format in their github repository, to facilitate reanalysis of the data.

We appreciate the Reviewer's attention to detail. Indeed, the initial dataset uploaded on Github was subset of the master dataset, while the code itself was for program testing. During revision, we have uploaded the complete dataset as MS Excel sheet in the GitHub repository. This document is machine-readable and should facilitate reanalysis of the data.

11. In a similar vein, the code on their github repository appears to be an example, rather than code that can be used to replicate the results shown in the paper. Again, including this would facilitate reproducing their work.

We have now presented a clean and detailed version of the codes with necessary instructions, including version numbers of various packages for necessary dependencies. All this should allow for smooth run of the codes and reproducibility of our work.

12. The code uses relatively standard python packages. However, for reproducibility, the authors should provide version numbers for the packages that they use (seabird, tensor flow, sci-kit, gpflow, etc.). Depending on the particular python package manager they are using, they should provide a configuration file which specifies the necessary dependencies and version information (e.g., setup.yaml or environment.yml file)

This question is addressed together with Comment #11 by Reviewer#3. **Please refer to our previous response.**

13. It would be helpful to clarify in the data availability statement on p.24 that the provided link <https://doi.org/10.5281/zenodo.1030139> only contains the ecological impact data.

We thank the Reviewer for this suggestion. During the revision, **we also uploaded the complete dataset of our experiments and that needed for running the active learning program in machine readable format to the Zenodo repository.**(<https://doi.org/10.5281/zenodo.11117242>).

(Remarks on code availability):

14. As noted in our comments for the authors, the code and data provided appear to be a sample or demonstration, rather than the actual data/code used for the study.

This question is addressed together with Comment #11 by Reviewer#3. **Please refer to our previous response.**

15. Additionally, they can improve the instructions for installation (and improve reproducibility) by providing version information, as noted in our comments.

This question is addressed together with Comment #12 by Reviewer#3. **Please refer to our previous response.**

Reviewer #4

We thank the Reviewer for co-reviewing our work. We have taken due care to address all the suggestions by the Reviewers regarding experimental and computational aspects, to improve the quality and presentation of our study.

REVIEWERS' COMMENTS

Reviewer #1 (Remarks to the Author):

The authors have addressed all my concerns. I now recommend acceptance of the manuscript.

Reviewer #2 (Remarks to the Author):

The authors have made sufficient revision. The paper is now acceptable.

Reviewer #3 (Remarks to the Author):

Overall, the manuscript has improved significantly. The clarity of some definitions and data availability has been enhanced, but there are a few minor points that still require attention before publication:

1. Line 541, equation (1) lacks one right bracket

2. Line 549, what is the chosen parameter? We recommend clarifying the hyperparameter choices/optimization by stating in the text that this is described in Supplementary Note 7. (We knew this only from reading the response to other reviewer comments, but a naive reader would not easily find it.)

3. Line 610, "step(v)" should be "step(iv)"

4. In the github: <https://github.com/ssuvarnamanu/active-learning-for-HAS?tab=readme-ov-file>, the authors write a text description of the package versions, in response to our early question. However, their current description is incomplete. For example, the Jupyter notebook "Active_learning_Phase_2_model.ipynb" needs to install `wheel` package. To avoid this type of problem and improve reproducibility, we recommend including all package versions in the python environment in a machine-readable way. For example, using `pip list` can extract the environment dependency information or using `pip freeze > requirements.txt` can export the information to a file. The resulting requirements.txt file can be put into the repository for use by other users.

5. Additionally, this file has some small code errors, for example, `Y=np.array(Y).reshape(-1, 1)` should be added before scale Y since the scaling function requires 2-dimension data. We encourage the authors to confirm that this (and other scripts) run correctly.

6. Although the authors have included additional data and essential parts of code, the code for reproducing the figures in the paper are not provided, so the presented results are not entirely replicable. We defer to the Editor if this level of reproducibility is a requirement for publication.

Reviewer #3 (Remarks on code availability):

See comments 4, 5, 6 in the Remarks to Authors, above.

Reviewer #4 (Remarks to the Author):

Response to Reviewers and Editorial requests for Manuscript NCOMMS-24-14026A

Code: Comments in **blue** | Replies in black | Actions in **bold**

Note: Indicated page, line, figure, or reference numbers refer to the revised manuscript and/or supplemental information with changes highlighted.

Reviewer #1

The authors have addressed all my concerns. I now recommend acceptance of the manuscript.

We thank the Reviewer for their valuable suggestions and constructive feedback that helped us to further improve our study, and for supporting its publication in *Nature Communications*.

Reviewer #2

The authors have made sufficient revision. The paper is now acceptable.

We thank the Reviewer for supporting our work and recommending for publication in *Nature Communications*.

Reviewer #3

Overall, the manuscript has improved significantly. The clarity of some definitions and data availability has been enhanced, but there are a few minor points that still require attention before publication:

We thank the Reviewer for recognizing the improved clarity and depth of our work, and their attention to detail in the critical feedback provided is duly appreciated. We have carefully revised the manuscript to address the remaining minor issues, as follows.

1. Line 541, equation (1) lacks one right bracket.

We have corrected this typographical error in Equation 7 (page 21, line 515).

2. Line 549, what is the chosen parameter? We recommend clarifying the hyperparameter choices/optimization by stating in the text that this is described in Supplementary Note 7. (We knew this only from reading the response to other reviewer comments, but a naive reader would not easily find it.)

We thank the Reviewer for raising this unclear point. **We have clarified in the revised manuscript that the hyperparameter choices/optimization are described in Supplementary Note 7 (page 21, lines 525-527).**

3. Line 610, “step(v)” should be “step(iv)”

We have corrected this typographical error (page 23, line 578).

4. In the github: <https://github.com/ssuvarnamanu/active-learning-for-HAS?tab=readme-ov-file>, the authors write a text description of the package versions, in response to our early question. However, their current description is incomplete. For example, the Jupyter notebook “Active_learning_Phase_2_model.ipynb” needs to install `wheel` package. To avoid this type of problem and improve reproducibility, we recommend including all package versions in the python environment in a machine-readable way. For example, using `pip list` can extract the environment dependency information or using `pip freeze > requirements.txt` can export the information to a file. The resulting requirements.txt file can be put into the repository for use by other users.

We thank the Reviewer for their useful suggestion. **We have provided an additional file containing all the library specific dependencies in GitHub.**

5. Additionally, this file has some small code errors, for example, `Y=np.array(Y).reshape(-1, 1)` should be added before scale Y since the scaling function requires 2-dimension data. We encourage the authors to confirm that this (and other scripts) run correctly.

We thank the Reviewer for this suggestion and **have amended the code and uploaded a revised notebook ‘Active_learning_Phase_2_model.ipynb’ directly in GitHub. All codes have been tested and they operate as expected.**

6. Although the authors have included additional data and essential parts of code, the code for reproducing the figures in the paper are not provided, so the presented results are not entirely replicable. We defer to the Editor if this level of reproducibility is a requirement for publication.

All the figures presented in the manuscript have been prepared using the graphing software Origin Pro. To facilitate reproducibility of figures, all the relevant figure data is provided as an excel sheet titled Source File. Additionally, we have uploaded the Origin Pro files to the Zenodo repository (<https://doi.org/10.5281/zenodo.11113829>); which should allow reproducibility of figures.

(Remarks on code availability):

See comments 4, 5, 6 in the Remarks to Authors, above.

This question is addressed together with Comments #4-6 by Reviewer#3. **Please refer to our previous responses.**

Reviewer #4

We thank the Reviewer for co-reviewing our work. We have taken due care to address all the suggestions by the Reviewers to improve the quality and presentation of our study.